# A new look at the architecture and dynamics of the *Hydra* nerve net

Athina Keramidioti[1†], Sandra Schneid[1†], Christina Busse[1†],
Christoph Cramer von Laue[2], Bianca Bertulat[2], Willi Salvenmoser[3], Martin Hess[1],
Olga Alexandrova[1], Kristine M Glauber[4], Robert E Steele[4], Bert Hobmayer[3*],
Thomas W Holstein[2*], Charles N David[1*]

[1]Department of Biology, Ludwig-Maximilians-University Munich, Martinsried, Germany; [2]Centre for Organismal Studies (COS) Heidelberg, Heidelberg University, Heidelberg, Germany; [3]Department of Zoology and Center for Molecular Biosciences Innsbruck (CMBI), University of Innsbruck, Innsbruck, Austria; [4]Department of Biological Chemistry, University of California, Irvine, United States

*For correspondence:
bert.hobmayer@uibk.ac.at (BH);
thomas.holstein@cos.uni-heidelberg.de (TWH);
david@bio.lmu.de (CND)

†These authors contributed equally to this work

Competing interest: The authors declare that no competing interests exist.

**Abstract** The *Hydra* nervous system is the paradigm of a 'simple nerve net'. Nerve cells in *Hydra*, as in many cnidarian polyps, are organized in a nerve net extending throughout the body column. This nerve net is required for control of spontaneous behavior: elimination of nerve cells leads to polyps that do not move and are incapable of capturing and ingesting prey (Campbell, 1976). We have re-examined the structure of the *Hydra* nerve net by immunostaining fixed polyps with a novel antibody that stains all nerve cells in *Hydra*. Confocal imaging shows that there are two distinct nerve nets, one in the ectoderm and one in the endoderm, with the unexpected absence of nerve cells in the endoderm of the tentacles. The nerve nets in the ectoderm and endoderm do not contact each other. High-resolution TEM (transmission electron microscopy) and serial block face SEM (scanning electron microscopy) show that the nerve nets consist of bundles of parallel over-lapping neurites. Results from transgenic lines show that neurite bundles include different neural circuits and hence that neurites in bundles require circuit-specific recognition. Nerve cell-specific innexins indicate that gap junctions can provide this specificity. The occurrence of bundles of neurites supports a model for continuous growth and differentiation of the nerve net by lateral addition of new nerve cells to the existing net. This model was confirmed by tracking newly differentiated nerve cells.

## eLife assessment

This work presents **important** findings on the cellular and ultrastructural organization of the nervous system in the freshwater polyp Hydra. The authors present outstanding imaging data with **convincing** evidence to support their claims. The manuscript provides a starting point for further functional in vivo studies. The work will be of interest to developmental biologists and neurobiologists.

## Introduction

Nerve cells arose early in the evolution of multicellular animals to sense the environment and control the activity of downstream effector cells (*Anderson, 1990*; *Mackie, 1990*). In non-bilaterians (cnidarians and ctenophores) nerve cells are organized in nerve nets, which are often associated with muscle processes and coordinate the behaviors that involve ectodermal and endodermal tissues (*Mackie, 1990*). In bilaterian metazoans, the number of nerve cells is commonly much higher than in cnidarians

and ctenophores and – importantly – nerves are organized into a centralized nervous system with large ganglia (*Arendt et al., 2016*). The relatively simple organization of nervous tissue in non-bilaterians offers the possibility of achieving a comprehensive understanding of the relationship between the architecture and activity of the nervous system and animal behavior (*Bosch et al., 2017*; *Galliot et al., 2009*; *Watanabe et al., 2009*).

With a nervous system consisting of only 500–2000 nerve cells, depending on animal size (*Bode et al., 1973*), the freshwater cnidarian polyp *Hydra* is a particularly useful model organism for such investigations. Furthermore, there are only two types of nerve cells in *Hydra* based on morphology: ganglion cells and sensory cells, and there are no surrounding support cells, such as glial cells in bilaterians (*David, 1973*; *Epp and Tardent, 1978*; *Westfall, 1973*; *Westfall and Epp, 1985*). Ganglion cells are bipolar or multipolar, but there is no evidence for a distinction between axons and dendrites. Sensory nerve cells, by comparison, are polarized, with a short neurite and sensory cilium at one pole and a longer neurite at the opposite pole. Sensory cells in the ectoderm extend a cilium to the surface of the epithelium to sense the external environment. Sensory cells in the endoderm extend a cilium into the gastric cavity. Recent scRNAseq results have refined these morphological classes to include three sensory and five ganglion nerve cell populations based on distinct transcriptomes (*Cazet et al., 2023*; *Siebert et al., 2019*).

Nerve cells in *Hydra* are organized in a nerve net, as first described in classic studies (*Schneider, 1890*; *Hadži, 1909*), in which the nerves were visualized by methylene blue staining and tissue maceration. Although higher concentrations of nerve cells were found in the head and the foot, there was no evidence for ganglia or a 'brain'. A more detailed description of the organization and formation of distinct subpopulations of nerve cells in the various body parts of *Hydra* polyps was achieved by using monoclonal antibodies that recognized unknown antigens and antibodies that recognized neuropeptides (*Dunne et al., 1985*; *Grimmelikhuijzen, 1983*; *Grimmelikhuijzen et al., 1996*; *Hobmayer et al., 1990b*; *Hobmayer et al., 1990a*; *Technau and Holstein, 1996*; *Yaross et al., 1986*). More recently, the use of in situ hybridization to detect the expression of neuropeptide genes and transgenic reporter lines using the promoters from neuropeptide genes has provided more understanding of the regionalization of the peptidergic part of *Hydra*'s nervous system (*Hansen et al., 2000*; *Grimmelikhuijzen et al., 1996*; *Koizumi et al., 2004*; *Noro et al., 2019*). However, all of these results lacked the resolution to describe the entire nerve net in detail and there is presently no clear understanding of the overall organization of cells in the nerve net or how the cells communicate with each other. Likewise, while extensive EM studies have provided high-resolution images of individual nerve cells, neurites, and synapses (*Kinnamon and Westfall, 1981*; *Kinnamon and Westfall, 1982*; *Westfall, 1973*; *Westfall et al., 1971*; *Westfall et al., 1983*), they could not reveal the overall organization of the nerve net.

The *Hydra* nerve net is unique in that it grows continuously. All *Hydra* tissues, including the nerve net, are newly formed every 2–3 weeks in well-fed animals (*Campbell, 1967*; *David and Campbell, 1972*; *David and Gierer, 1974*). Since cell proliferation occurs exclusively in the body column, there is continuous displacement of cells from the body column toward the head and foot and loss of cells at the extremities. In the course of this growth and displacement, the nerve net expands by the addition of nerve cells, which arise by differentiation from interstitial stem cells in the body column. Head- and foot- specific nerve cells are also continually added to these differentiated tissues. For example, tentacle tissue is completely renewed in 4–5 days by displacement of epithelial tissue from the body column into the tentacles. Ectodermal epithelial cells at the base of a tentacle differentiate into battery cells with the incorporation of nematocytes and newly differentiated sensory nerve cells (*Hobmayer et al., 1990b*).

In the present experiments, we take advantage of a newly developed pan-neuronal antibody (PNab), which stains all nerve cells and nerve processes in fixed *Hydra* polyps. Antibody staining and confocal microscopy show that nerve cells in *Hydra* are organized as two separate nerve nets, one in the ectoderm and one in the endoderm. Transmission electron microscopy (TEM) and block face scanning electron microscopy (SEM) demonstrate that the two nerve nets consist of discrete bundles of overlapping nerve cell processes. The bundles include different nerve cell types and neural circuits. Cell type-specific expression of innexins and formation of gap junctions provides the necessary circuit specificity. Strikingly, these neurite bundles extend past nerve cell bodies and do not appear to terminate on them. These results suggest a novel model for growth of the nerve net based on the lateral

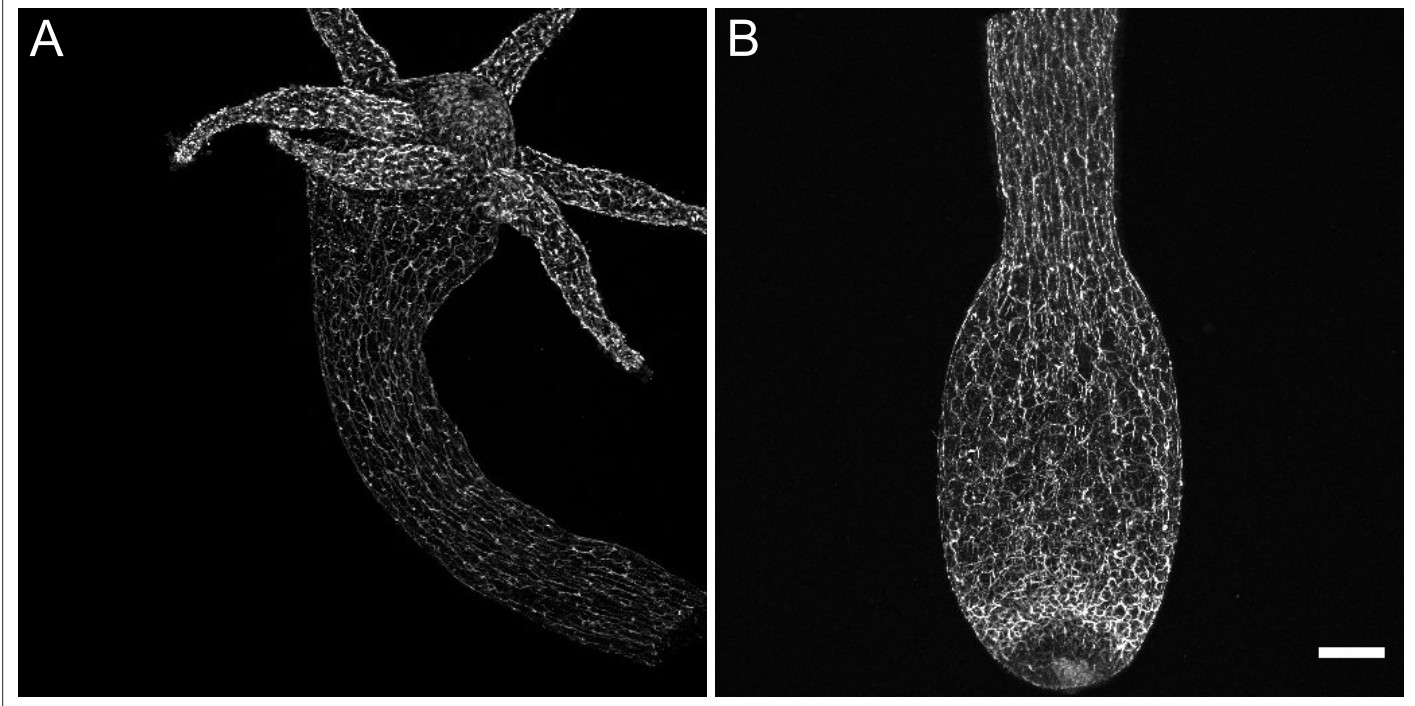

**Figure 1.** Distribution of pan-neuronal antibody (PNab)-stained nerve cells along the *Hydra* body axis. The confocal images are maximum intensity projections of short stacks (10–15 µm) showing the ectoderm. (**A**) Upper body column including the hypostome and tentacles. (**B**) Lower body column including the peduncle and basal disk. Scale bar: 150 µm.

The online version of this article includes the following figure supplement(s) for figure 1:

**Figure supplement 1.** Structure of *Hydra* tissue imaged in transgenic polyps expressing DsRed2 in ectodermal epithelial cells and GFP in endodermal epithelial cells.

**Figure supplement 2.** Representative images of pan-neuronal antibody (PNab)-stained body column of a transgenic watermelon polyp (GFP in ectoderm, DsRed2 in endoderm) used to count epithelial cells and nerve cells.

addition of new nerve cells to the existing net. We confirmed this model by tracking the addition of newly differentiated nerve cells to the nerve net.

## Results

### A new peptide antibody stains the entire nerve net in *Hydra*

A rabbit antibody made against the peptide sequence VTRNQQDQQENRFSNQ (see Materials and methods) stains the nerve net in *Hydra* (**Figure 1**) and is referred to here as PNab. In order to determine whether PNab recognizes all nerve cells in *Hydra*, two sets of experiments were performed. First, we investigated transgenic strains of *Hydra*, which express NeonGreen or GFP in nerve cells. One transgenic strain uses the promoter of an alpha-tubulin gene driving expression of NeonGreen. This transgene has been demonstrated to be strongly expressed in all differentiated nerve cells in *Hydra*, and correspondingly the entire nervous system is brightly NeonGreen-positive (**Primack et al., 2023**). Fixed polyps of this strain were stained with PNab and transgenic NeonGreen-expressing nerve cells were scored for PNab staining (see Materials and methods). All NeonGreen-expressing nerve cells (684/684) were stained with PNab.

We also scored smaller numbers of PNab-stained nerve cells in two additional transgenic lines, i.e., Hym176B (**Noro et al., 2019**) and nGreen (see Materials and methods). In the Hym176B line 149/149 transgenic nerve cells were stained with PNab. In the nGreen line 51/51 transgenic nerve cells were stained with PNab. We conclude from these results that PNab stains all nerve cells in *Hydra* and hence permits imaging of the entire nerve net.

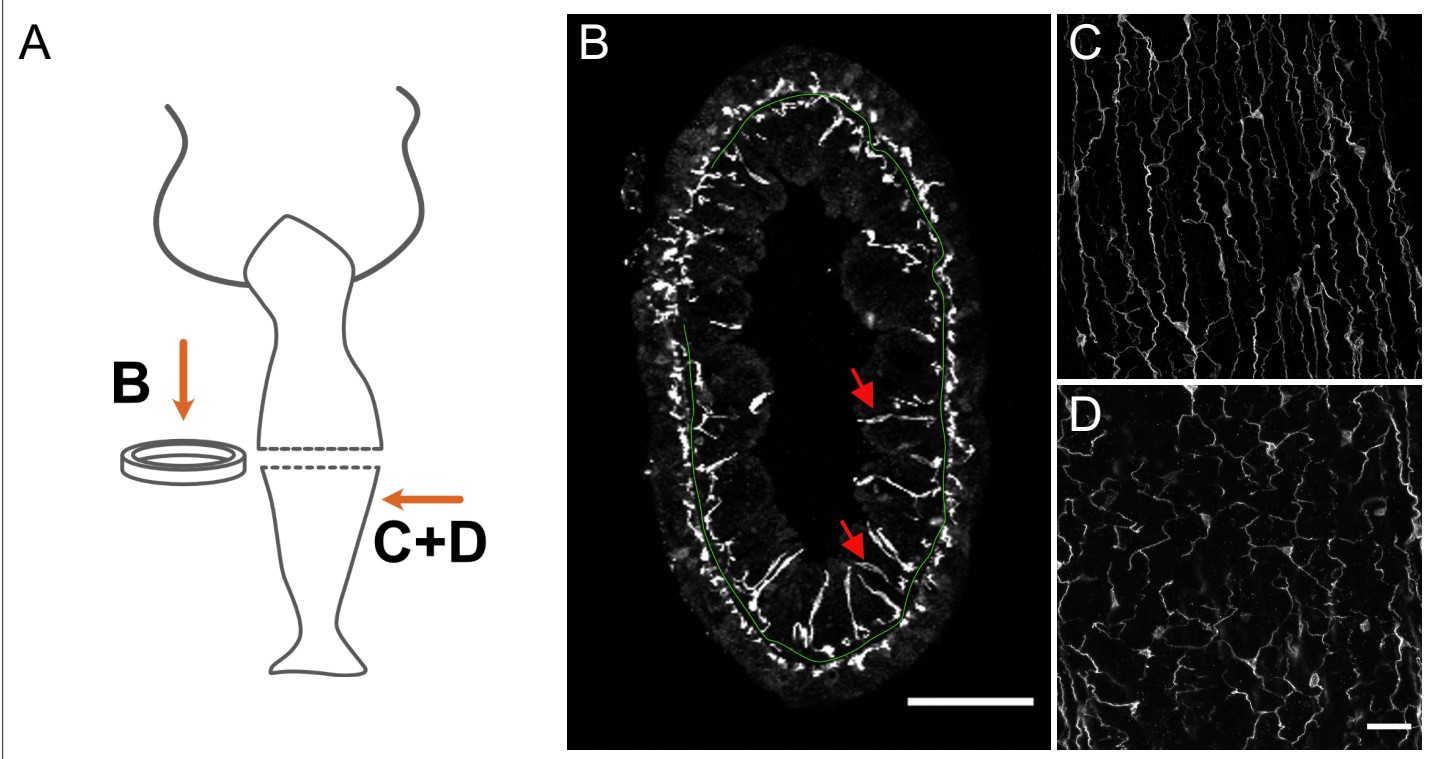

**Figure 2.** Distribution of pan-neuronal antibody (PNab)-stained nerve cells in a cross-section of the body column. (**A**) Schematic shows the position of imaged tissue. (**B**) Circular ring of tissue excised from the body column and viewed from above. The green line traces the position of the mesoglea separating ectoderm and endoderm except where tissue is compressed in the upper left quadrant. The ectodermal and endodermal nerve nets are separate structures. Red arrows indicate sensory cells extending from the endodermal ganglion net into the gastric cavity. (**C** and **D**) Maximum intensity projection images of short stacks through the body wall showing parallel tracks of neurites and nerve cell bodies in the ectoderm (**C**) and a polygonal pattern of nerve cells and neurites at the same position in the endoderm (**D**). Oral is to the top, aboral to the bottom. Scale bars: (**B**) 100 μm, (**C** and **D**) 30 μm.

In a second set of experiments we compared the number of PNab-stained nerve cells in confocal image stacks with the number of nerve cells determined by maceration of the same tissue (see Materials and methods). Both measurements yielded a ratio of 0.12 nerve cells/epithelial cell (*Figure 1—figure supplement 2*) confirming that PNab is a pan-neuronal marker staining all nerve cells in *Hydra*.

### Imaging the *Hydra* nerve net with PNab

*Figure 1A* shows a low-magnification confocal image of the upper body column of *Hydra* stained with PNab, including the hypostome and tentacles. The image shows primarily the ectodermal nerve net; endodermal nerve cells lie below the focal plane. The antibody stains both nerve cell bodies and nerve cell processes, which are oriented parallel to the oral-aboral axis of the body column and also extend into the tentacles. The hypostome forms a dome above the tentacle ring and includes nerve cells oriented toward the mouth opening in the center of the hypostome. *Figure 1B* shows a confocal image of the lower half of a body column. Ectodermal nerve processes are oriented along the oral-aboral axis but become less oriented in the peduncle. There is an increase in nerve cell numbers in a band just above the basal disk. Confocal images of 'inverse watermelon' polyps expressing DsRed2 and GFP in ectodermal and endodermal epithelial cells, respectively, show the tissue structure in which the nerve net is embedded (*Figure 1—figure supplement 1*).

*Figure 2A and B* shows a ring cut from the body column in the mid-gastric region with the ectodermal and endodermal nerve nets. The view is down the oral-aboral axis and the position of the mesoglea is shown as a green line for orientation (see also *Figure 1—figure supplement 1C*). The ectodermal nerve net consists of ganglion cells oriented along the oral-aboral axis and hence cut in cross-section. The endodermal nerve net consists of roughly equal numbers of sensory cells and ganglion cells. The ganglion cells form a network near the base of endodermal cells whereas the

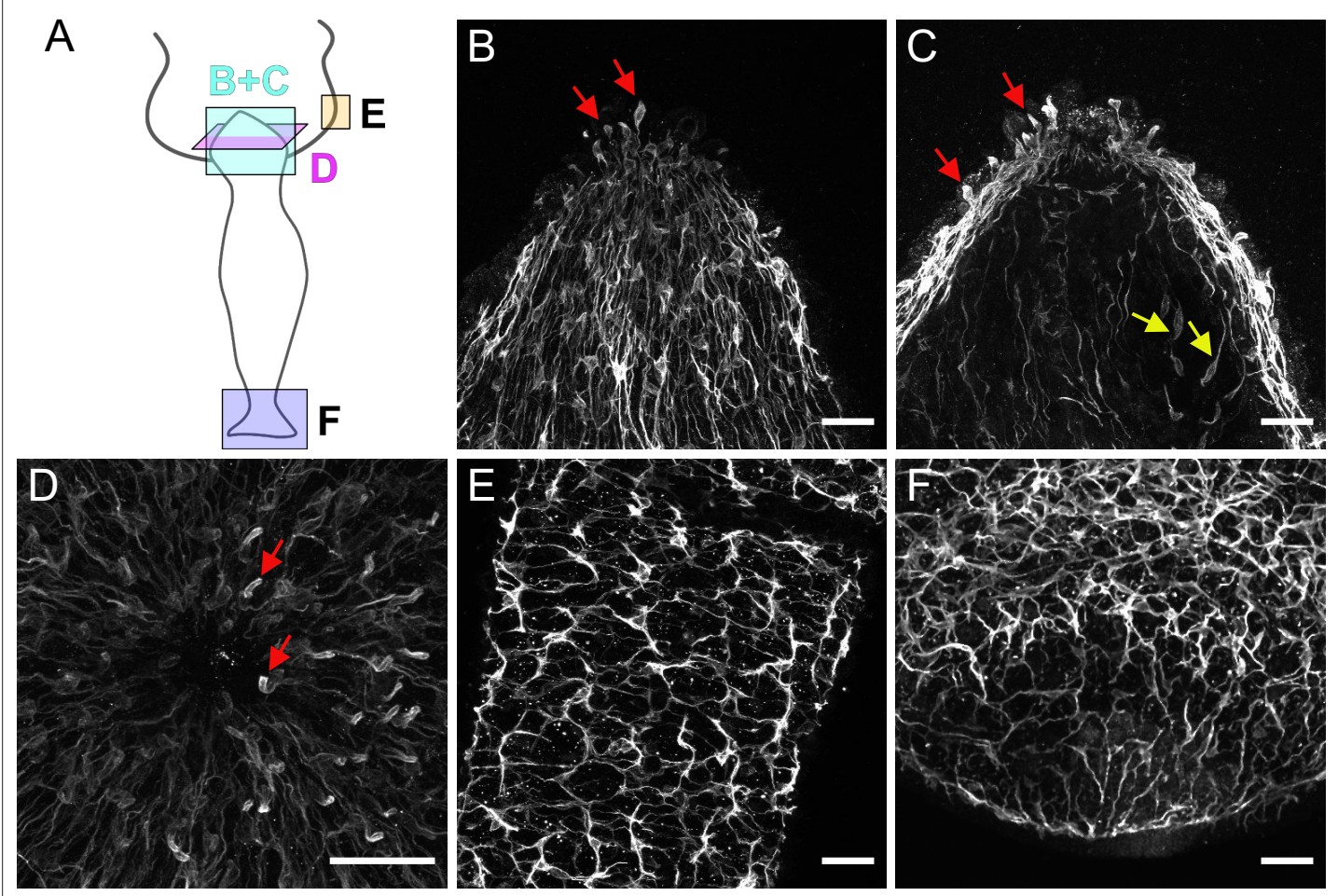

**Figure 3.** Pan-neuronal antibody (PNab)-stained nerve cells in the hypostome (**B**, **C**, **D**), tentacle (**E**), and peduncle/basal disk (**F**). (**A**) Schematic shows the orientation of the confocal images. The images are maximum intensity projections of short stacks (10–15 µm) through the ectoderm of the hypostome (**B**) and deeper in the tissue at the same position (**C**) showing sensory nerve cells in the ectoderm (red arrows) and in the endoderm (yellow arrows). (**D**) Ectoderm surrounding the mouth opening viewed from above showing sensory nerve cells in ectoderm (red arrows). (**E**) Ectodermal nerve net in a short section of tentacle tissue. (**F**) Ectodermal nerve net in the peduncle and basal disk. Scale bars: (**B**, **C**, **D**, **E**, and **F**) 30 µm.

sensory cells extend prominently up through the endodermal layer into the gastric cavity. *Figure 2C and D* shows images from a single confocal stack taken through the body wall in the gastric region. Nerve cells in the ectoderm form an array of parallel processes oriented along the oral-aboral axis (*Figure 2C*); those in the endoderm form a roughly polygonal array (*Figure 2D*). Both nerve cell bodies and nerve cell processes stain with the PNab. The nerve cell bodies are roughly 10 µm long; the distance between nerve cell bodies is variable in both ectoderm and endoderm but usually 50–100 µm. While cell bodies and processes in the ectoderm are organized in a plane, which includes the muscle processes of the ectodermal cells, those in the endoderm are not closely associated with endodermal muscle processes (see below). Endodermal sensory cells extend deeper into the tissue (see *Figure 2B*) and are not included in the image.

Higher magnification images (*Figure 3*) of the hypostome, a tentacle, and the peduncle/basal disk reveal specific neuronal patterns with a higher density of nerve cells. *Figure 3B* is a surface view of the hypostome showing a dense network of ganglion cells with nerve processes extending down the body column. *Figure 3C* shows a section deeper in the hypostome roughly passing through the mouth opening. Sensory cells are prominent in the ectoderm near the mouth and extend to the outer surface of the tissue (*Figure 3B and C*, red arrows). When viewed from above, these sensory nerve cells form a ring around the mouth opening (*Figure 3D*, red arrows). Sensory cells in the endoderm (*Figure 3C*, yellow arrows) lie between the dense mass of mucous cells and endodermal epithelial cells, which

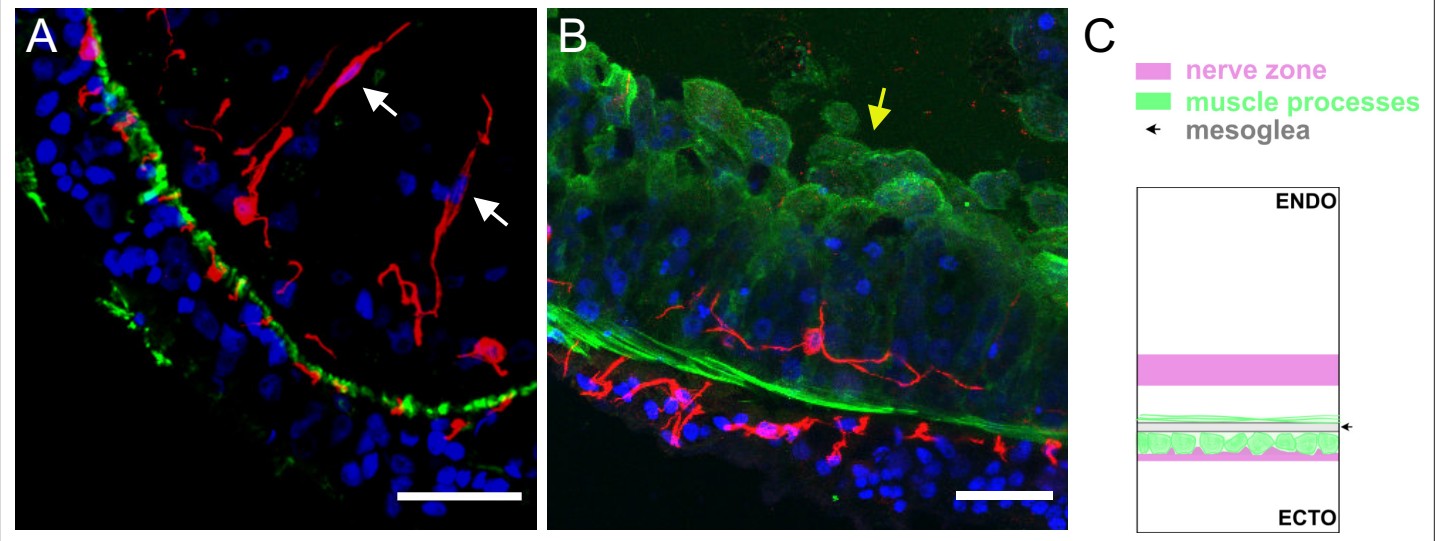

**Figure 4.** Segments of a ring of body column tissue excised perpendicular to the main body axis from Lifeact ecto (**A**) and Lifeact endo (**B**) polyps stained with pan-neuronal antibody (PNab) (red) and anti-GFP (green). The confocal images are maximum intensity projections of short stacks (20 µm) down the body column. (**A**) Lifeact ecto polyp. In the ectoderm small red spots (neurites) and green spots (muscle processes) alternate and are directly adjacent to the unstained mesoglea. In the endoderm neurites of ganglion cells are located at some distance from the mesoglea. Two sensory nerve cells (white arrows) in the endoderm extend upward to the gastric cavity. (**B**) Lifeact endo polyp. Green muscle processes extend laterally in the plane of the image. The apical actin network of the endodermal cells faces the gastric cavity and is also strongly stained green (yellow arrow). A single ganglion cell in the endoderm extends red neurites parallel to, but separate from green muscle processes. (**C**) Schematic shows the position of ectodermal and endodermal nerve nets relative to corresponding muscle processes and the mesoglea. Scale bars: 30 µm.

make up the endodermal layer of the hypostome (see *Figure 1—figure supplement 1A and B*). They are connected to a network of ganglion cells in the endoderm under the mesoglea.

*Figure 3E* shows a surface view of a portion of an extended tentacle. Nerve cells form a roughly polygonal pattern associated with the battery cells, which form the ectoderm in tentacles (see *Figure 1—figure supplement 1D*). Single sensory nerve cells (Nv-1 positive; *Hobmayer et al., 1990b*) are integrated within battery cell complexes (*Cramer von Laue, 2003*; *Hobmayer et al., 1990b*). In the peduncle (*Figure 3F*) there is a broad band of increased nerve cell density, which is formed by a unique population of unusually large ganglion nerve cells. These cells stain strongly with PNab and have been shown to express the gap junction protein innexin 2 (*Takaku et al., 2014*), and the neuro-peptides Hym 176A and RFa-A (*Noro et al., 2019*).

## The nerve nets in the ectoderm and endoderm are separate structures and are differently associated with muscle processes

While the images in *Figures 1–3* describe the complete nerve net, they do not permit localization of nerve cells or portions of the nerve net clearly to the ectoderm or the endoderm. To resolve this question, we used transgenic lines expressing Lifeact-GFP (*Aufschnaiter et al., 2017*), which binds specifically to filamentous actin (*Riedl et al., 2008*). Since most filamentous actin is concentrated in the muscle processes at the base of epithelial cells, these structures are brightly fluorescent in confocal images. The muscle processes are localized at the base of epithelial cells directly apposed to the mesoglea. Thus, they precisely mark the boundary between ectoderm and mesoglea and between endoderm and mesoglea.

*Figure 4* shows segments of rings of body column tissue (see *Figure 2A*) cut from an ectodermal Lifeact animal (*Figure 4A*) and from an endodermal Lifeact animal (*Figure 4B*). In *Figure 4A* the green spots represent muscle processes in the ectoderm, which run parallel to the oral-aboral axis of *Hydra*, and were cut when the ring was excised from the body column. The irregular red dots in between muscle processes in the ectoderm are nerve cell processes. They are typically closely associated with muscle processes in the ectoderm. Underneath the ectodermal muscle processes there is a black

space representing the mesoglea, which separates the ectoderm from the endoderm. Red PNab-stained nerve cell bodies and neurites are also present in the endoderm.

*Figure 4B* shows the opposite situation in an endodermal Lifeact animal. Muscle processes at the base of endodermal epithelial cells are oriented circumferentially around the body column and mark the endodermal side of the mesoglea. An endodermal ganglion cell is stained by PNab and neurites from this cell extend parallel to but are separated from the muscle fibers. Thus, there is a clear difference in the way the ectodermal and endodermal nerve nets are associated with muscle processes. The diagram in *Figure 4C* shows this schematically. The localization of ectodermal ganglion cells completely overlaps the localization of ectodermal muscle processes. By contrast, the localization of endodermal ganglion cells is largely separated from endodermal muscle processes.

Importantly, in *Figure 4A and B* there are no connections between nerve cells in the ectoderm and nerve cells in the endoderm. To search more extensively for nerve cell contacts between the ectoderm and endoderm, we imaged complete rings of body column tissue (see *Figure 2B*) from four different PNab-stained Lifeact polyps. These rings contained a total of 133 nerve cells (57 ectoderm,

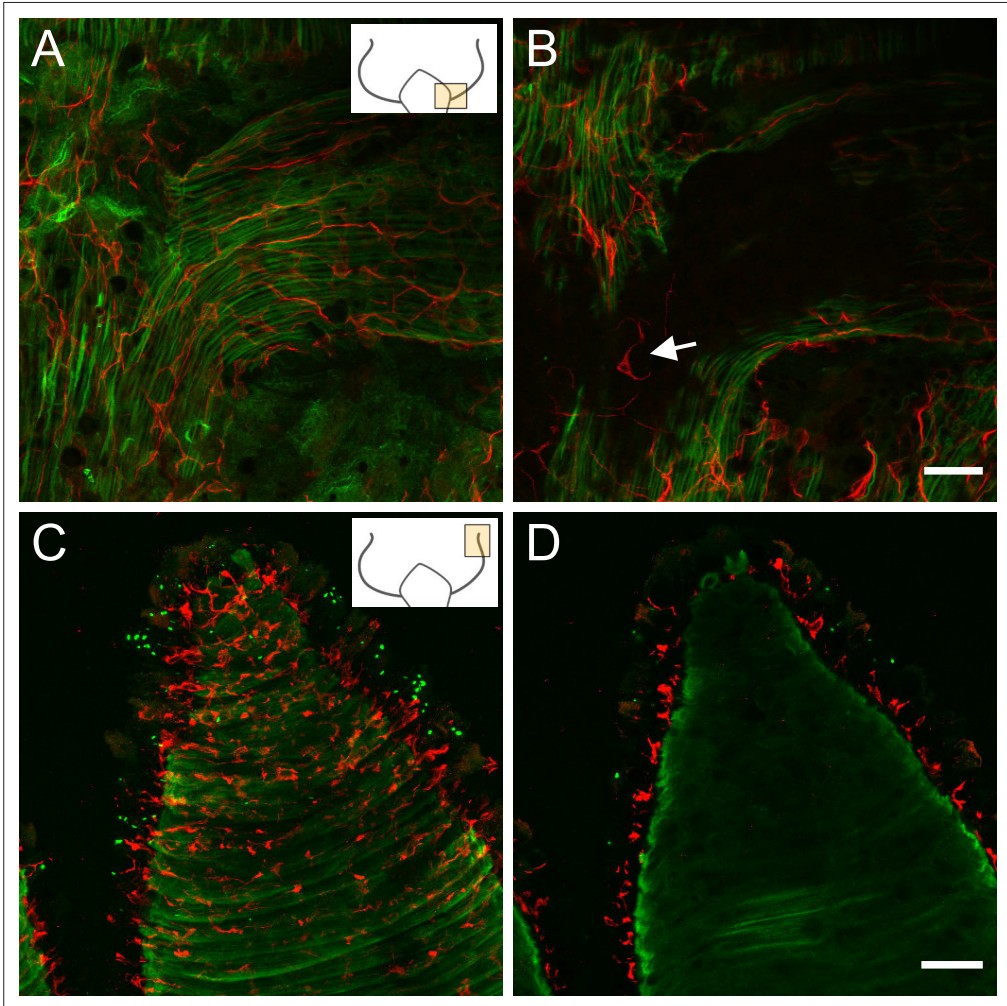

**Figure 5.** Lifeact ecto (**A** and **B**) and Lifeact endo (**C** and **D**) polyps stained with pan-neuronal antibody (PNab). Confocal images are maximum intensity projections of image stacks in the ectoderm and endoderm. Small schematics indicate position of the images. (**A**) Lifeact ecto polyp showing the base of a tentacle. Green muscle processes extend parallel to the oral-aboral body axis and into the base of the tentacle. Red neurites are closely associated with muscle processes and extend into the tentacle. (**B**) Image at the same position as (**A**) but deeper into the tissue showing absence of nerve cells in the endoderm of the tentacle but a nerve cell and neurites in the endoderm of body column (white arrow). (**C**) Lifeact endo polyp showing a tentacle surrounded by circular green muscle processes. Red nerve cells form a net in the ectoderm. (**D**) Same position but deeper in the tissue than (**C**), showing complete absence of red nerve cells in the endoderm. Scale bars: (**A** and **B**) 30 µm, (**C** and **D**) 100 µm.

76 endoderm) and associated neurites and represented about 4% of the body column tissue. No contacts between the ectodermal and endodermal nerve nets were observed. We have taken many more image stacks at different positions in the body column, peduncle, and hypostome and never found a connection between the ectodermal and endodermal nerve nets.

## There are no nerve cells in the endoderm of tentacles

In his original description of the *Hydra* nerve net, Hadzi noted that nerve cells were not present in the endoderm of tentacles (*Hadži, 1909*). We have confirmed this unexpected result using PNab staining of Lifeact animals.

*Figure 5A* shows an image of the ectoderm at the base of the tentacles in an ectodermal Lifeact animal. The PNab-stained nerve net lies above the green muscle processes of the ectodermal epithelial cells. The muscle processes are oriented parallel to the oral-aboral axis of the polyp and extend into the base of the tentacle. An image taken deeper in the tissue, below the ectodermal muscle processes (*Figure 5B*), reveals PNab-stained nerve cells in the endoderm of the body column but no nerve cells in the endoderm of the tentacles. We also made image stacks through tentacles of endodermal Lifeact animals stained with PNab (*Figure 5C and D*). There are many red nerve cells in the ectoderm above the green muscle processes (*Figure 5C*), but no nerve cells in the endoderm below the muscle processes (*Figure 5D*).

The absence of neurons in the tentacle endoderm indicates a massive loss of endodermal nerve cells during the formation of tentacle tissue from tissue of the upper body column. We suggest that this loss of endodermal neurons is related to the formation of battery cells in the ectoderm (*Hobmayer et al., 1990b*) and reflects a remodeling of both the ectoderm and endoderm during tentacle formation.

## Nerve cell numbers in the ectoderm and endoderm along the body column

Images of the nerve net in *Figures 1–3* show only small differences in the apparent density of nerve cells along the body column. To put this qualitative observation on a quantitative basis, we counted nerve cells and epithelial cells in confocal image stacks through the body wall at various positions along the body column from the hypostome and tentacles to the basal disk. The results are shown in *Figure 6*. Most strikingly, there were no nerve cells in the endoderm of tentacles (see above). Similarly, the basal disk had very few nerve cells in the endoderm. In contrast to these two differentiated structures, the density of nerve cells in the body column endoderm is relatively constant (0.10–0.19) from the hypostome to the peduncle. The density of nerve cells in the ectoderm, however, changes dramatically along the body column. While the ratio of nerve to epithelial cells is relatively constant (0.15–0.22) in the gastric region and the peduncle, the number of nerve cells in the ectoderm rises sharply in the hypostome, tentacles, and basal disk. These are regions known to have a high level of new nerve cell differentiation (see Discussion).

## The nerve net consists of closely associated parallel tracks of neurites (bundles)

The images in *Figures 1–3* do not reveal how the nerve net is constructed, i.e., how the size and shape of individual nerve cells contribute to the overall structure of the net. Do the connections between nerve cells in the net consist of overlapping neurites from neighboring cells or single neurites extending from one cell and synapsing on the next cell? To acquire such information, we stained transgenic nGreen polyps with PNab. This strain is a mosaic, in which not all stem cell precursors are transgenic. As a consequence, in a population of nGreen animals some polyps contain more and some fewer GFP-labeled nerve cells. We took advantage of this mosaicism to identify animals with relatively few GFP-labeled nerve cells. This allowed us to find isolated GFP-labeled nerve cells and thus image how such an individual nerve cell contributes to the PNab-stained nerve net.

*Figure 7* shows two GFP-labeled nerve cells in the ectoderm with long neurites extending parallel to the body axis. The nGreen neurites (*Figure 7A*) are so closely associated with red PNab-stained neurites (*Figure 7B*) that it is almost impossible to distinguish the two. Only occasionally were the green or red neurites locally separated from each other so that both were visible (*Figure 7C*). This

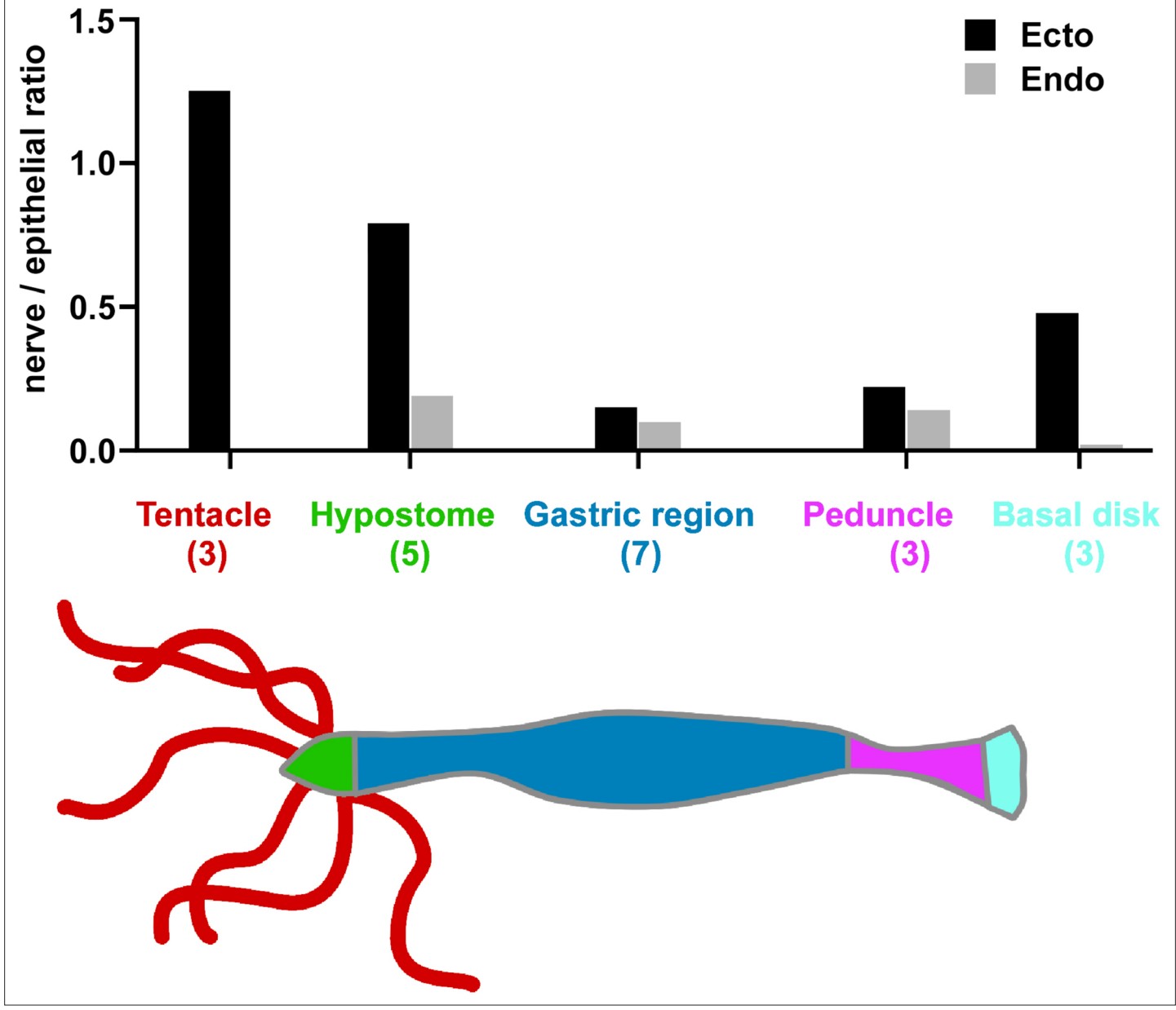

**Figure 6.** Nerve cell to epithelial cell ratio along the body column of *Hydra*. Transgenic watermelon and inverse watermelon polyps were stained with pan-neuronal antibody (PNab) to identify nerve cells and with DAPI to identify nuclei. Confocal stacks were taken through the body wall at various positions along the body column and scored for ectodermal and endodermal epithelial cells and for nerve cells (see Materials and methods and *Figure 1—figure supplement 2*). The results are semi-quantitative, since the number of animals scored (shown in brackets) and the number of image stacks varied at each position. A total of 80–400 nerve cells were scored in body column and peduncle samples, 20–30 in hypostome samples, but only 0–2 in the endoderm of tentacle and basal disk due to absence or very rare occurrence of endodermal nerve cells in these body parts.

showed that over long stretches the nerve net consists of bundles of neurites running parallel to each other.

The occurrence of bundles of parallel neurites suggests that a neurite from one cell may not only extend along the neurite of a neighboring cell, but also past the cell body of that neighboring cell. We have found numerous examples of such cell bodies in high-resolution confocal images. *Figure 8* shows an example in which a red PNab-stained neurite passes next to a green GFP-labeled nerve cell body (*Figure 8A–C*). A single optical section (*Figure 8D–F*) shows clearly that both the GFP-labeled and unlabeled neurites are stained with PNab. We also found examples in PNab-stained preparations in which several tracks of neurites passed over a nerve cell body. In contrast to the above situation,

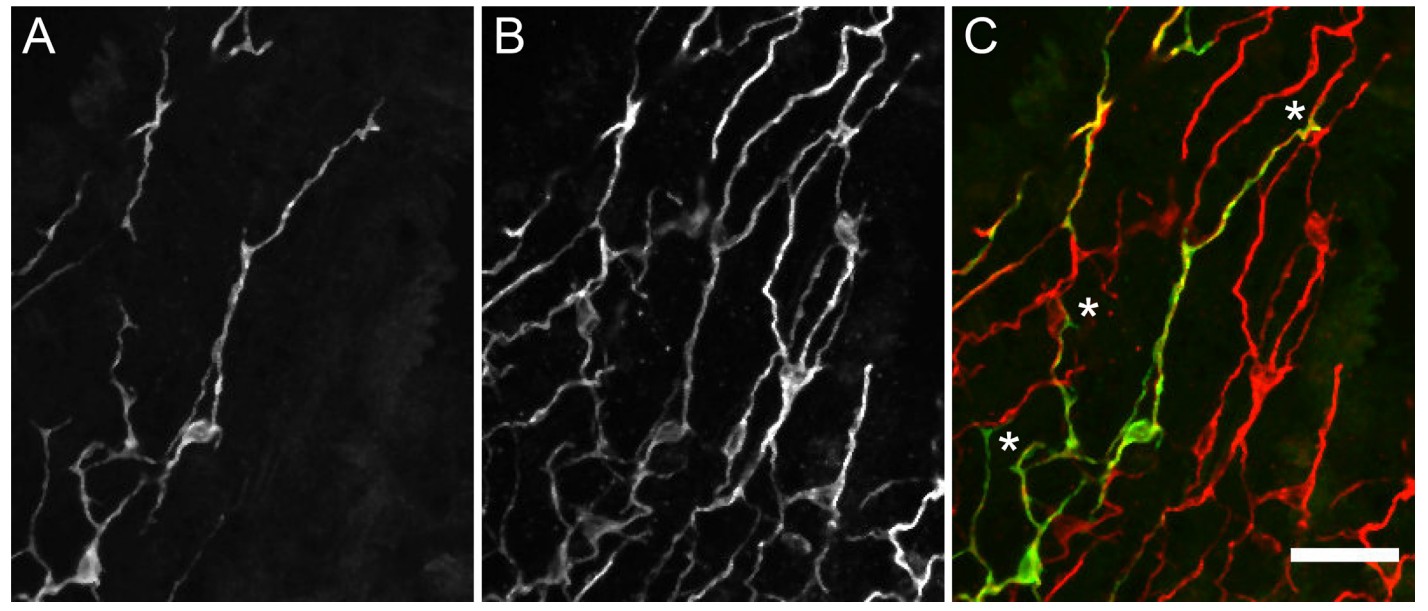

**Figure 7.** Transgenic nGreen polyp stained with anti-GFP (green) and pan-neuronal antibody (PNab) (red). Confocal images of the ectoderm in the body column showing two GFP-labeled nerve cells overlapping the PNab-stained nerve net. (**A**) Anti-GFP, (**B**) PNab, and (**C**) overlay of anti-GFP (green) and PNab (red) images. White asterisks mark three GFP-labeled neurites, which run parallel to and terminate on the PNab-stained nerve net. Note that GFP-labeled nerve cells are also stained with PNab. Scale bar: 30 µm.

we found few examples of neurites ending on the cell body of a neighboring cell. The ends of GFP-labeled neurites usually lie somewhere along the nerve net at a site that was otherwise unremarkable (see asterisks in *Figure 7C*). Thus, communication along the nerve net appears to be a case of cell-to-cell communication by neurites interacting laterally to transmit a signal.

## EM confirms the widespread occurrence of neurite bundles

We sought to confirm the unexpected and widespread occurrence of neurite bundles by TEM. *Hydra* polyps were fixed and stained for TEM and sectioned perpendicular to the body axis. *Figure 9A* is a low-magnification image of the ectoderm in the middle of the body column showing both the apical surface and the basal muscle fibers (green) of the epithelial cells lying directly over the mesoglea (yellow). The muscle processes are cut in cross-section and form a continuous mat immediately adjacent to the mesoglea. A bundle of four neurites lies next to a nerve cell body (nvb, red) and a second bundle with three neurites lies to the left of the nerve cell body. Both neurite bundles are directly adjacent to muscle processes in agreement with the PNab-stained image in *Figure 4A*. *Figure 9B* shows a higher magnification image of another bundle of four neurites in the ectoderm. The neurites contain numerous darkly stained glycogen granules and dense core vesicles, which are typically found in nerve cells (*Westfall, 1973*).

In addition, we used TEM cross-sections in the mid-body column to measure the frequency of neurite bundles with different numbers of neurites in a bundle. In the ectodermal nerve net, only about 20% of the cross-sectioned nerve tracks occurred as single neurites (*Figure 9C*). About 80% of the nerve tracks occurred as bundles, many as bundles of two neurites, but often also as bundles consisting of three to five neurites (*Figure 9C*).

*Figure 9D* shows a TEM cross-section of the endoderm in the middle of the body column with the basal muscle processes (green) directly adjacent to the mesoglea (yellow). Two major differences in nervous system architecture were obvious when endoderm was compared with ectoderm. First, endodermal neurites and neurite bundles, containing numerous dense core vesicles, were located at some distance from the endodermal muscle processes (*Figure 9D and D'*), equivalent to the findings obtained with PNab staining in confocal images (*Figure 4B*). Second, about half of the endodermal nerve tracks were single neurites, the other half were bundles containing two neurites (*Figure 9C*). Bundles with more than two neurites were not found in the endoderm. Thus, in summary, the

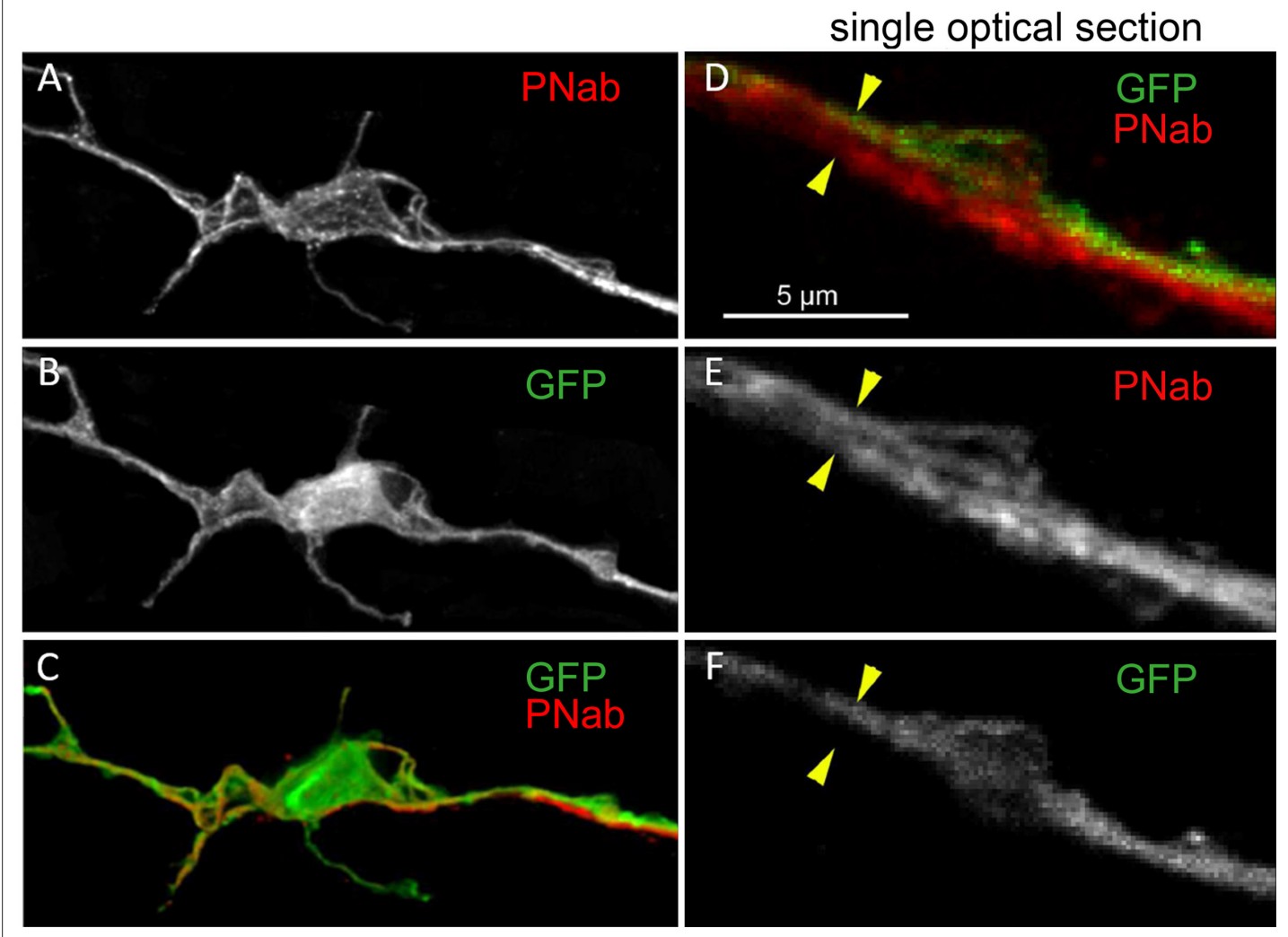

**Figure 8.** Transgenic nGreen polyp stained with pan-neuronal antibody (PNab) (red) and anti-GFP (green). (**A–C**) Maximum intensity projection confocal image stack of a GFP-stained nerve cell tightly associated with a PNab (red)-stained neurite from another cell. (**D–F**) Single optical section from the image stack in (**A**) showing parallel GFP and non-GFP neurites, both stained with PNab. The yellow arrowheads mark the outer boundary of the two parallel neurites in (**D**, **E**, and **F**). Scale bar: 5 µm.

ectodermal nerve net is more directly associated with muscle processes and clearly exhibits higher connectivity than the endodermal nerve net.

To extend the TEM results, we analyzed 2000 serial sections of the ectodermal epithelium in the mid-body region. Serial sectioning was carried out with the block face SEM method. *Figure 10A* shows a 3D reconstruction of the serial sections. The imaged block is 56×40×13 µm³ and includes two nerve cell bodies and corresponding parallel tracks of neurites (purple and green) lying adjacent to muscle processes of the ectodermal epithelial cells. Neurites from three additional nerve cells (turquoise, yellow, and red) whose cell bodies lie outside the sectioned block run parallel to and are closely associated with the neurites from the two nerve cells shown. The 3D reconstruction confirms at high resolution that nerve tracks between nerve cell bodies consist of bundles of two or more tightly associated nerve processes. The distance between the two nerve cell bodies along the oral-aboral axis is roughly 50 µm, in agreement with the distribution of nerve cell bodies shown in *Figure 2C*.

A striking feature of the reconstruction in *Figure 10A* are neurites which run past the two nerve cell bodies (blue, purple, and green). This result is in agreement with the images in *Figures 7–9A*, which also show neurites running past neighboring nerve cell bodies. While closely attached to these nerve cell bodies, the neurites do not appear to terminate on them. *Figure 10B and C* are views from above and below the reconstruction taken from an interactive 3D model showing only the nerve cells

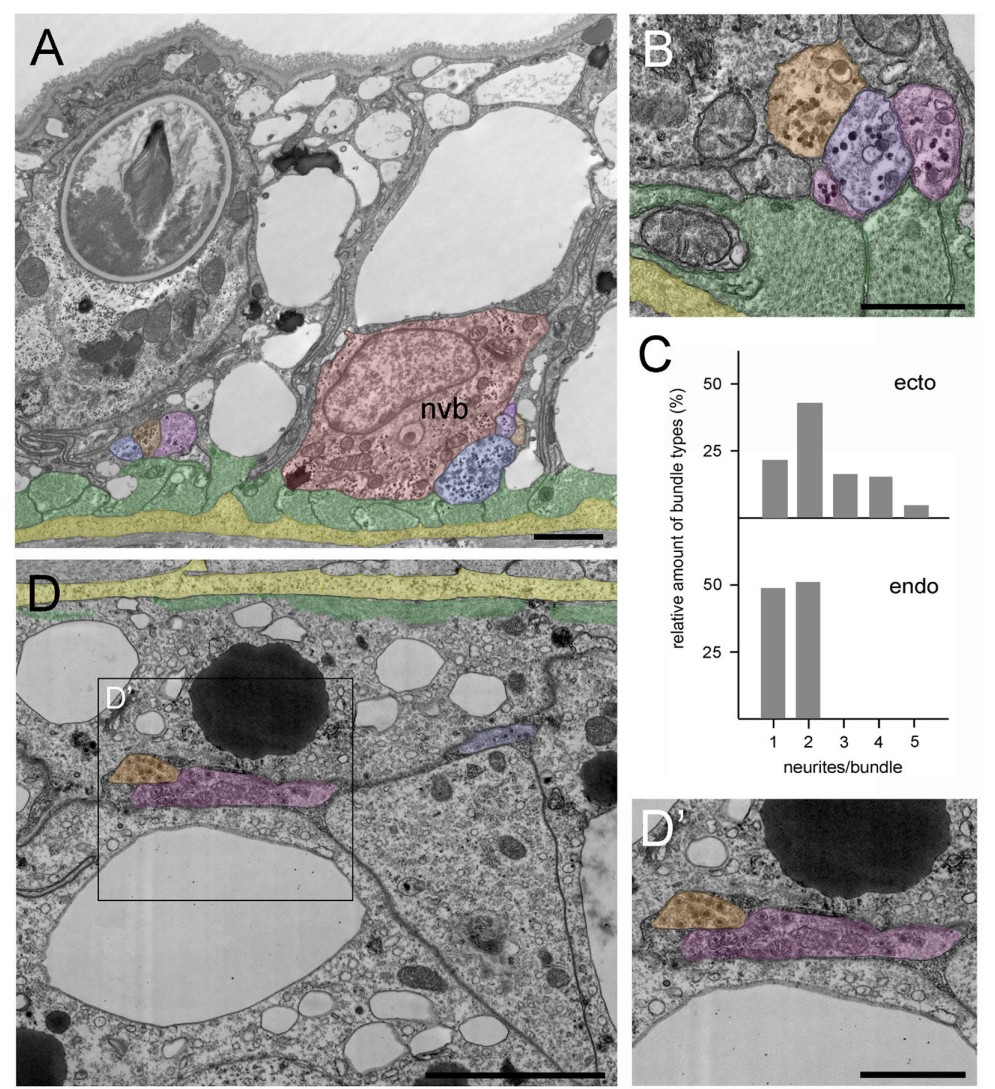

**Figure 9.** Position and structure of neurite bundles in the mid-body column in transmission electron microscopy (TEM) cross-sections. To facilitate identification, neurites are overlaid in orange-blue-purple, the nerve cell body in red, muscle processes in green, and the mesoglea in yellow. (**A**) Low magnification of the ectoderm. A nerve cell body (nvb) is directly associated with the muscle processes, and four neurites cut in cross-section run parallel to the nerve cell body. An additional neurite bundle containing three neurites with contact to a muscle process is visible to the left of the nerve cell body. (**B**) Higher magnification image of another bundle of four neurites in contact with and running parallel to muscle processes. (**C**) Quantitative analysis of the relative amount of neurite bundle complexity using four TEM cross-sections spanning 40 µm in the mid-body column. Analysis included 130 nerve tracks in the ectoderm and 47 nerve tracks in the corresponding endodermal area. In the ectoderm, about 20% of nerve tracks consist of single neurites, about 80% of bundles contain two to five neurites. In contrast, about 50% of endodermal nerve tracks consist of single neurites, and about 50% contain two neurites. (**D**) Cross-section of the endoderm showing a single neurite and a bundle with two neurites located at some distance to the basal muscle processes and the mesoglea. (**D'**) Higher magnification of the endodermal nerve bundle reveals numerous dense core vesicles and mitochondria. Scale bars: (**A**) 10 µm, (**B** and **D**) 5 µm, (**D'**) 2 µm.

and neurites (*Figure 10—source data 1*). The blue, green, and purple neurites extend over the entire length of the reconstructed block parallel to the long axis. By comparison, the yellow and red neurites meet in the middle and form thin terminal fingers, which overlap at the contact site for about 8 µm. The yellow neurite also forms a thin finger, which appears to contact a spur of the purple neurite for about 4 µm (*Figure 10C*). Thus there are two qualitatively different connections between neurites:

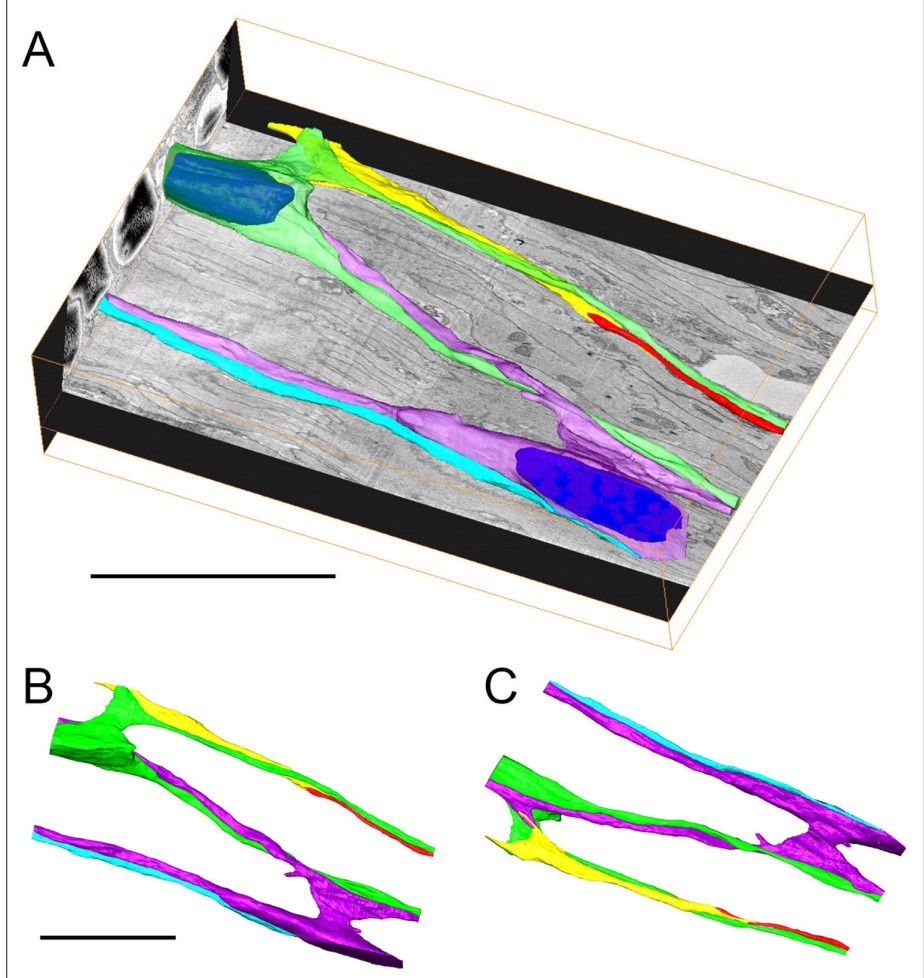

**Figure 10.** 3D reconstruction of 2000 serial sections obtained by serial block face scanning electron microscopy (SEM) in the ectoderm of the body column. (**A**) Two nerve cell bodies and their neurites are colored green and purple (nuclei are highlighted in blue). Three additional neurites from nerve cells outside the imaged block are colored yellow, red, and blue. The imaged block is 56 µm long. (**B** and **C**) Top and bottom projections from an interactive model of the reconstruction showing only nerve cell bodies and neurites (*Figure 10—source data 1*). Scale bars: 20 µm.

The online version of this article includes the following source data for figure 10:

**Source data 1.** 3D interactive model of the serial block face scanning electron microscopy (SEM) reconstruction shown in *Figure 10*.

short, thin, overlapping regions, which suggest contact, and long parallel tracks apparently without contact. The short spur of the purple neurite (*Figure 10C*) contacts a muscle process.

## Neurite bundles and neural circuits

The occurrence of neurite bundles raises questions of which nerve cell types and nerve circuits are involved. Based on the *Hydra* single cell atlas, there is only one ganglion nerve cell population in the endoderm (en1), and this population forms the endodermal nerve net (*Cazet et al., 2023*). En1 corresponds to the endodermal neural circuit RP2 (*Dupre and Yuste, 2017*). By comparison, there are two nerve cell populations (ec1/ec5 and ec3), which form the nerve net in the ectoderm. These two populations correspond to the contraction burst CB and RP1 neural circuits in the ectoderm (*Dupre and Yuste, 2017*; *Giez et al., 2023b*).

To visualize how ec1/ec5 and ec3 nerve cells contribute to nerve bundles in the ectodermal nerve net, we stained a Hym176B transgenic line (*Noro et al., 2019*), which expresses GFP in ec1 nerve

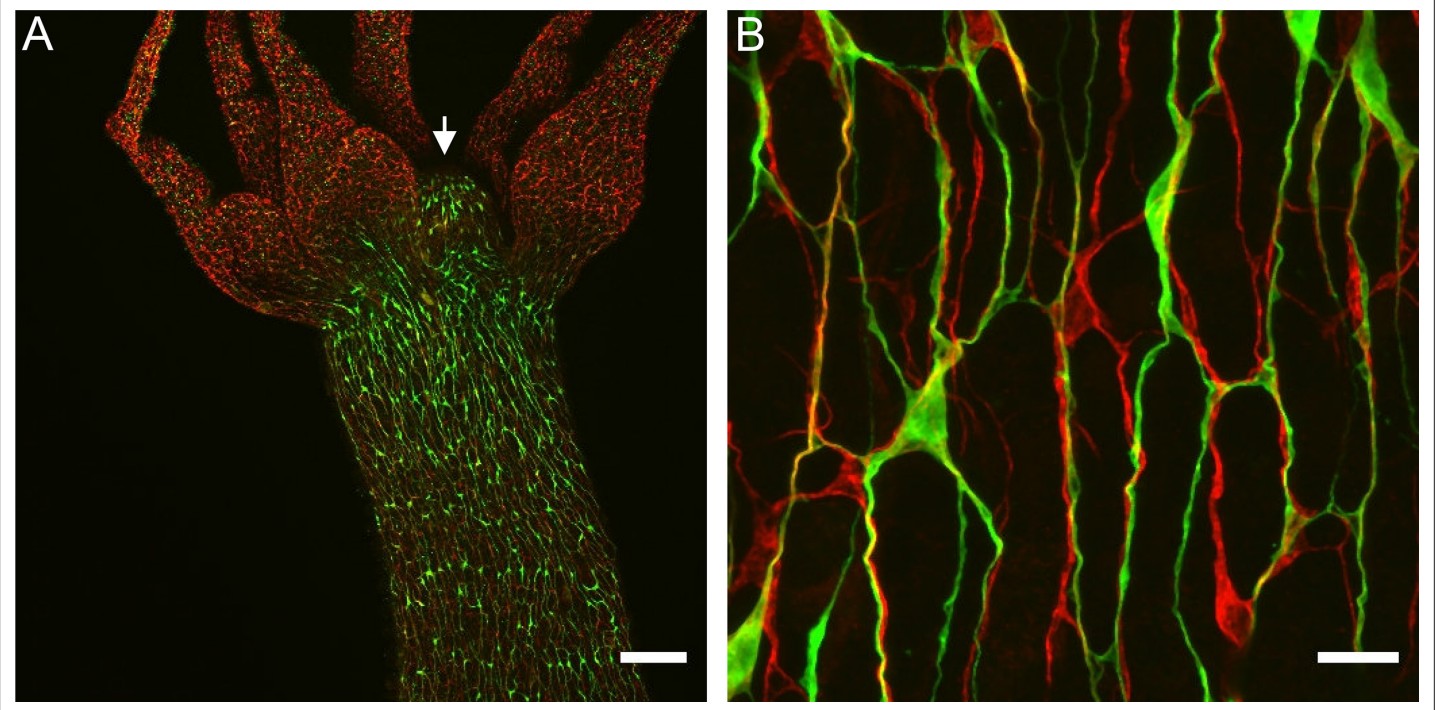

**Figure 11.** Nerve cell composition of neurite bundles. Transgenic Hym176B animal stained with anti-GFP and pan-neuronal antibody (PNab).
(**A**) Overview of hypostome (white arrow), tentacles, and upper body column showing transgenic Hym176B nerve cells stained with anti-GFP (green) and all nerve cells stained with PNab (red). (**B**) High-magnification image of the same Hym176B animal showing nerve cell bodies and neurite bundles in the nerve net. Most bundles consist of both Hym176B neurites and PNab-stained neurites (stained only with PNab). Scale bars: (**A**) 150 µm, (**B**) 15 µm.

cells in the hypostome and body column, with PNab, which stains all nerve cells. *Figure 11A* shows that Hym176B nerve cells form a continuous nerve net in the hypostome and body column. There are no Hym176B cells in the tentacles. The Hym176B cells are also stained with PNab. Parallel to the Hym176B nerve net there are additional nerve cells stained only with PNab. These are ec3 nerve cells. At higher magnification (*Figure 11B*) it is clear that both ec1 and ec3 nerve cells are closely associated in neurite bundles. Since these two ganglion nerve cell populations correspond to the two different neural circuits CB and RP1, neurites in the bundles need to form specific synaptic connections with the corresponding nerve cell type. As shown below, circuit-specific gap junctions provide the necessary specificity.

## Neurites in bundles are linked by circuit-specific gap junctions

The TEM and block face images in *Figures 9 and 10* do not have sufficient resolution to identify chemical synapses or gap junctions between neurites in bundles. To determine how neurites in bundles communicate with each other we used higher resolution TEM images on longitudinal sections of the ectoderm. *Figure 12A, B* shows two examples of gap junctions between neurites in bundles. Detecting gap junctions by block face SEM or TEM is however difficult and it is not possible to determine which neural circuits are involved. Alternatively, scRNAseq results can now be used to detect innexin transcripts and reveal which cell types contain them. Innexins have been shown to form gap junctions in *Hydra* (*Alexopoulos et al., 2004*; *Takaku et al., 2014*) and a search of an updated scRNAseq database for *Hydra* (*Cazet et al., 2023*) has revealed that the two nerve cell populations constituting the CB and RP1 circuits in the ectoderm (*Dupre and Yuste, 2017*), each contain distinct innexin transcripts. The corresponding UMAPs are shown in *Figure 12C*. Both ec1 and ec5 nerve cells, which form the CB neural circuit, express innexin 2; ec3 nerve cells, which form the RP1 neural circuit, express innexins 6 and 14 (*Figure 12C*). Hence circuit-specific gap junctions can be formed by innexins uniquely expressed in nerve cells of each circuit. Innexin 2 is also expressed in en1 nerve cells (*Figure 12C*), which form the RP2 circuit in the endoderm. There is, however, no cross-correlation

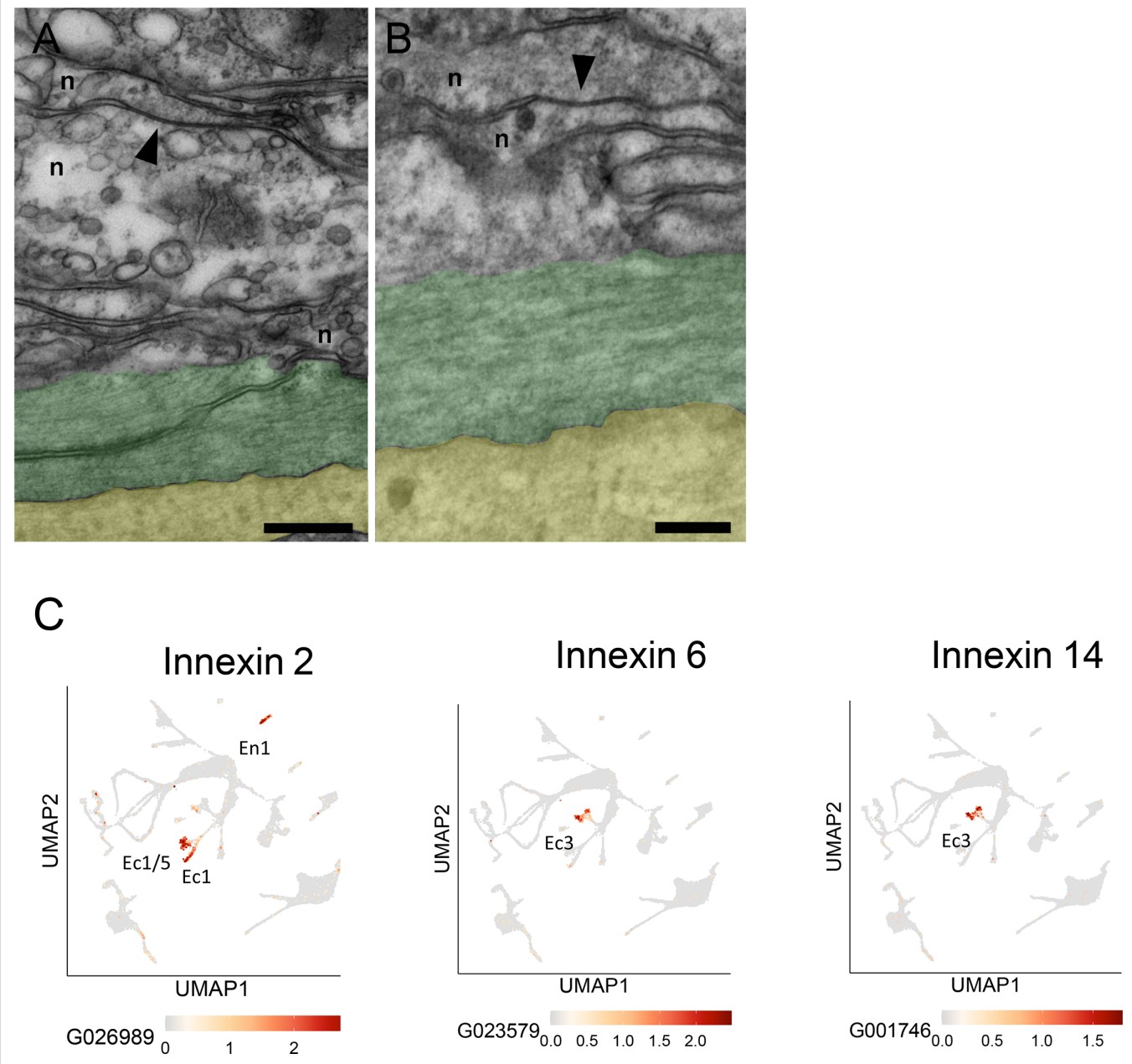

**Figure 12.** Gap junctions and innexins. (**A** and **B**) Transmission electron microscopy (TEM) images of longitudinal sections in the mid-body column of the ectoderm show neurite bundles running along muscle processes. Gap junctions connecting two neurites (n) are indicated by black arrowheads. To facilitate identification, muscle processes are overlaid in green and the mesoglea in yellow. Scale bar: 2 μm. (**C**) *Innexin* genes 2, 6, and 14 are uniquely expressed in specific nerve cell populations. UMAPs show expression (orange spots) of individual *innexin* genes in specific nerve cell populations: *innexin2* in Ec1/5 and Ec1 in the ectoderm and En1 in the endoderm; *innexin6* and *innexin14* in Ec3 in the ectoderm. The UMAP identifying cell populations is shown as a gray background. Data from *Cazet et al., 2023*. See https://biowebprod22.nhgri.nih.gov/HydraAEP/SingleCellBrowser/.

between the neural activity profiles of CB and RP2 and RP1 and RP2, indicating that the endodermal and ectodermal neural circuits are not connected (*Dupre and Yuste, 2017*).

## Nerve net growth occurs by lateral addition of new nerve cells to bundles

A unique feature of the *Hydra* nerve net is that it is a dynamic structure. It grows and is modified continuously. Since adult polyps do not increase in size, excess tissue generated by cell division is

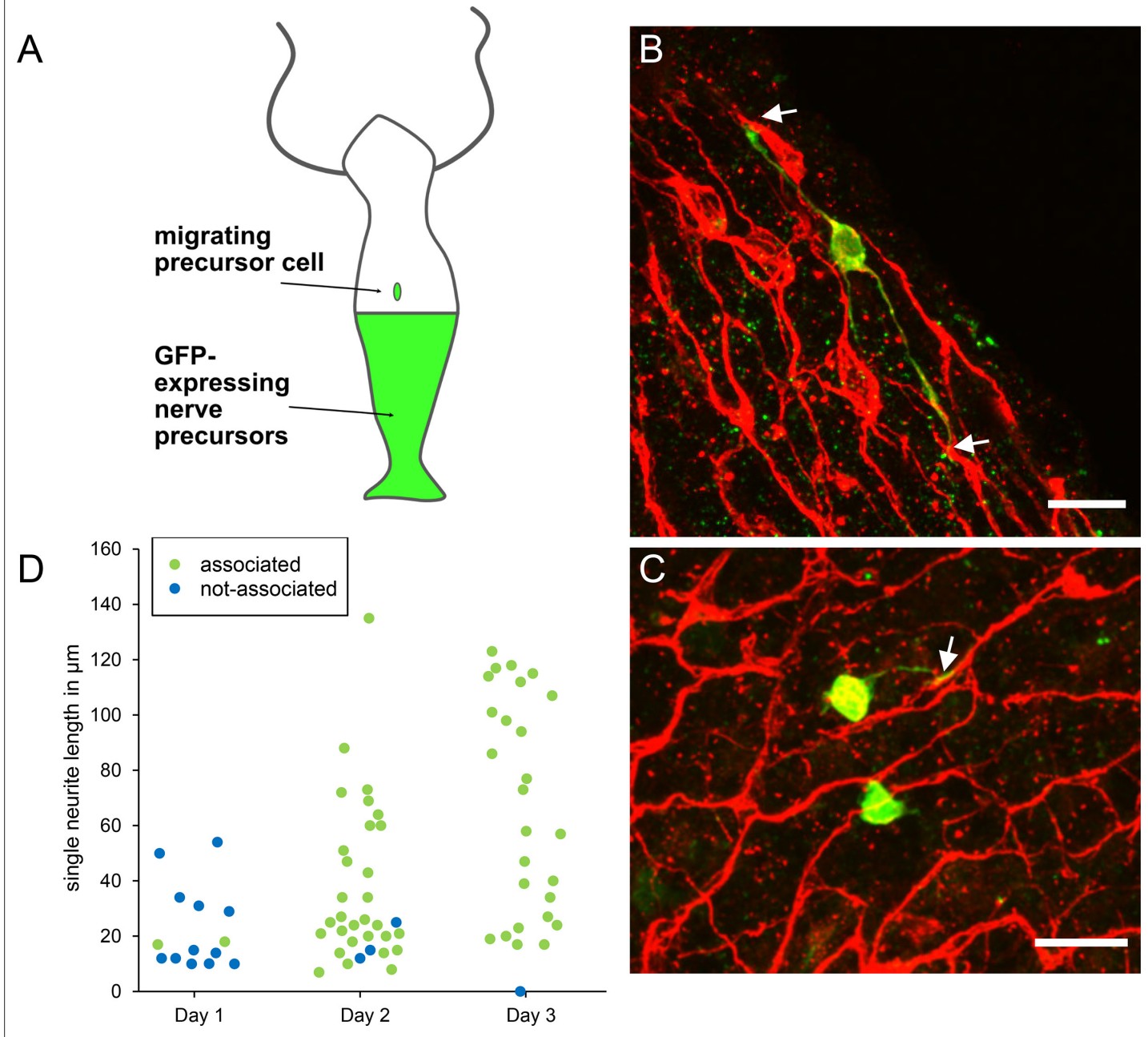

**Figure 13.** Differentiation of migrating nerve cell precursors in AEP (top)/nGreen (bottom) grafts stained with pan-neuronal antibody (PNab). (**A**) Schematic of AEP/nGreen graft. (**B** and **C**) Two images of differentiating nerve cell precursors (nGreen) on day 2 after grafting. Nerve net stained with PNab (red). White arrows mark association of nGreen neurites with the red nerve net. Differentiating nGreen nerve cells are stained with PNab and hence appear yellowish in the images. (**D**) Neurite outgrowth and association with the nerve net on days 1–3 after grafting. Each spot represents one nGreen nerve cell. Scale bars: 15 μm.

displaced into buds and to the ends of the body column (*Campbell, 1967*). Tentacle tissue is continuously renewed by differentiation of battery cells at the base of tentacles. Basal disk tissue is similarly renewed. Cells are lost by sloughing from the tips of the tentacles and the center of the basal disk. How new nerve cells are added to the nerve net was not known. The occurrence of bundles, however, suggests a simple way to add nerve cells to the net, by lateral association. Indeed, the image in *Figure 7* of two GFP-labeled nerve cells laterally associated with the PNab-stained nerve net supports a model of nerve net growth by lateral addition.

To investigate how new nerve precursors are added to the nerve net, we grafted unlabeled AEP polyps to transgenic nGreen polyps (*Figure 13A*). GFP-labeled nerve cell precursors migrated from the bottom half to the top half of such grafts and differentiated into new nerve cells. To localize the sites of new differentiation relative to the nerve net, we stained the grafts with PNab on days 1, 2, and 3 after grafting. *Figure 13B, C* shows images of young nerve cells (nGreen) associating with the nerve net (red) on day 2. Migrating precursors without neurites do not associate with the nerve net and most differentiating nerve cells with short neurites on day 1 were also not associated with the nerve net (*Figure 13D*). However, by days 2 and 3 most differentiating nerve cells extended neurites along the nerve net (*Figure 13D*). The results suggest that outgrowing neurites actively seek lateral associations with the neurites of the existing nerve net.

## Discussion

### There are two separate nerve nets in *Hydra*: implications for motor control and contractile activity

PNab staining has permitted, for the first time, detailed imaging of the entire nerve net in *Hydra*. The results reveal two separate nerve nets, a net of interconnected nerve cells spread throughout the ectoderm and a second nerve net in the endoderm. There are no nerve cells in the endoderm of tentacles (*Figure 5*) and very few in the endoderm of the basal disk (*Figure 6*). No connections between the two nerve nets were found based on PNab staining of ectodermal and endodermal nerve cells in rings cut from the body column of four animals (see Results). This is in agreement with earlier TEM observations (*Hufnagel and Kass-Simon, 1976*). It also agrees with neural activity measurements showing no cross-correlation between firing patterns of the ectodermal CB and RP1 neural circuits and the endodermal RP2 neural circuit (*Dupre and Yuste, 2017*).

In the ectoderm, nerve cell bodies and nerve cell processes are closely associated with muscle processes at the base of ectodermal epithelial cells (*Figure 4A and C*). TEM images of cross-sections of the body column confirm the close association of nerve cells with muscle processes in the ectoderm (*Figures 9 and 10*). This close association is consistent with direct neural control of body column contractions (see below). The localization of the endodermal nerve net is quite different. Endodermal nerve cells and their processes are located in a diffuse band at some distance from the endodermal muscle processes (*Figure 4B, C*; *Figure 9D*). Furthermore, the polygonal pattern of endodermal nerve cells (*Figure 2D*) is completely different from the parallel array of endodermal muscle processes arranged circumferentially around the body column (*Figure 4B*). Both results suggest that the endodermal nerve net is not directly involved in controlling the activity of endodermal muscle processes.

Recently, *Dupre and Yuste, 2017*, described electrical activity in nerve cells in a transgenic strain of *Hydra* that expresses the calcium reporter GCaMP6s in nerve cells (*Dupre and Yuste, 2017*). They identified three discrete non-overlapping populations of nerve cells based on activity profiles (CB, RP1, RP2). Cells in each population fired together and there was no overlap between the firing patterns. The RP2 population was located in the endoderm, while the RP1 and CB populations were located in the ectoderm. The activity of the CB network was directly correlated with rapid contraction of the body column. The activity of the RP1 and RP2 networks was associated with slower changes in behavior.

The RP2 population is identical with the endodermal nerve net identified by PNab staining (*Figure 2D*; *Figure 4B*). Moreover, there are no RP2 nerve cells in the tentacles, in agreement with the absence of endodermal nerve cells in the tentacles as assayed by PNab staining (*Figure 5*). The observation that the RP2 nerve net is not associated with rapid changes in behavior is consistent with the limited contact between the endodermal nerve net and endodermal muscle processes (*Figures 4B and 9D*). However, endodermal ganglion cells do strongly express several neuropeptides, e.g., GLWa (*Cazet et al., 2023*; *Siebert et al., 2019*) and release of such neuropeptides could modulate contractile activity in endodermal muscle processes. In particular *Takahashi et al., 2003*, have shown that GLWa induces slow contraction of the endoderm and consequently elongation of the body column (*Takahashi et al., 2003*).

The two ectodermal circuits (RP1 and CB) (*Dupre and Yuste, 2017*) correspond to the ectodermal nerve net identified by PNab staining (*Figures 2C and 4A*). Although the nerve net in the ectoderm appears to be homogeneous, the results of scRNAseq experiments indicate that it consists of

two populations of ganglion cells, which express distinct neuropeptides (*Cazet et al., 2023*; *Siebert et al., 2019*). Ec1 and Ec1/5 express the neuropeptide Hym176 (*Noro et al., 2019*) while Ec3 nerve cells express the neuropeptide GLWa.

To correlate neural activity with muscle activity, transgenic *Hydra* that express distinct calcium reporters in ectodermal and ectodermal epithelial cells were made (*Szymanski and Yuste, 2019*). This enabled detection of muscle activity in each epithelium during fast contraction of the body column initiated by CB nerve cells. The results showed that both ectodermal and endodermal epithelial cells responded simultaneously to activity of the CB circuit. Since there are no nerve cell connections between the ectoderm and the endoderm (see above), they postulated (*Wang et al., 2023*) that the ectodermal epitheliomuscle cells activate endodermal epitheliomuscle cells directly via gap junction connections across the mesoglea (*Hand and Gobel, 1972*). This conclusion is consistent with our findings. Nerve cells in the ectoderm are closely associated with ectodermal muscle processes (*Figures 4A, 9, and 10*) and TEM studies have described both chemical synapses and gap junctions between nerve cells and ectodermal muscle processes (*Westfall, 1973*; *Westfall et al., 1971*; *Westfall et al., 1980*). By comparison, our results indicate that such contacts are rare in the endoderm (*Figures 4B and 9D*) and there is no endodermal neural activity associated with the fast contraction of the body column (*Dupre and Yuste, 2017*).

## The *Hydra* nerve nets consist of bundles of parallel nerve processes linked by circuit-specific gap junctions

Prior to the present results, it was not clear how the nerve net in *Hydra* is constructed. One possibility was that nerve processes from neighboring cells simply meet and synapse 'end-to-end' or that a neurite from one cell synapses on the cell body of a neighboring nerve cell, as is the case in bilaterian nervous systems. *Figures 7 and 8* show that, surprisingly, this is not the case. The connections between nerve cells along the body column mainly consist of two or more neurites attaching laterally to form bundles in the ectoderm (*Figure 9C*). The EM images in *Figures 9 and 10* confirm that neurites in such bundles are closely associated and show in addition that neurites in bundles extend past neighboring nerve cell bodies. Furthermore, the images of the nerve net in transgenic Hym176B polyps (*Figure 11*) demonstrate that neurites in bundles in the ectoderm belong to two different neural circuits CB and RP1. Hence, nerve cell-specific connections are required to carry circuit-specific signals. Circuit-specific gap junctions between neurites in bundles (*Figure 12A, B, and C*) appear to provide the necessary specificity. Furthermore, this result is consistent with the observation of *Dupre and Yuste, 2017*, that nerve cells in the CB, RP1, and RP2 circuits fire simultaneously indicating that they are linked by gap junctions.

Earlier TEM studies have identified both chemical synapses and electrical synapses (gap junctions) between nerve cells and between nerve cells and epithelial cells in *Hydra* (*Kinnamon and Westfall, 1981*; *Kinnamon and Westfall, 1982*; *Westfall et al., 1971*; *Westfall et al., 1980*). These synapses occurred at multiple sites along neurites, so-called en passant synapses (*Westfall et al., 1971*), but were not associated with neurite bundles. The only previously described occurrence of neurite bundles was a 'nerve ring' surrounding the hypostome in *Hydra oligactis* (*Koizumi, 2007*). This nerve ring contained 20–30 nerve cells (*Koizumi et al., 1992*) and TEM images of the nerve ring (*Matsuno and Kageyama, 1984*) showed that neurites and nerve cell bodies were tightly associated, similar to what is seen in the images in *Figures 9 and 10*.

## Nerve cells and polyp form

The body column of *Hydra* is a tube formed by two sheets of epithelial cells separated by a basement membrane (mesoglea). The tube is closed at the basal end by the basal disk and at the apical end by the mouth. The epithelial cells have muscle processes abutting the mesoglea. The muscle processes of endodermal cells are arranged circumferentially around the body column (*Figure 4B*); muscle processes of ectodermal cells are arranged longitudinally, parallel to the oral/aboral axis (*Figure 4A*). The combined force of these two muscle sheets gives rise to the tubular shape of the body column.

In addition to muscle tension, the shape and behavior of the body column is affected by osmotic pressure. *Hydra* polyps live in fresh water and there is a constant influx of water into the cytoplasm of the epithelial cells. This water is transferred to the gastric cavity and periodically voided through the

mouth (*Benos and Prusch, 1972*; *Benos and Prusch, 1973*; *Yamamoto and Yuste, 2020*) and potentially through the pore in the center of the basal disc (*Shimizu et al., 2007*).

The nerve cells in the two epithelial layers affect the dynamic behavior of this system. This is clearly seen in nerve-free *Hydra* (*Campbell, 1976*; *Marcum and Campbell, 1978*). These animals no longer contract and are nearly motionless due to the absence of ectodermal nerve cells. The body column of nerve-free animals is also distended by increased osmotic pressure, which cannot be released through the closed mouth. However, regular experimental opening of the closed mouth in nerve-free animals leads to release of the gastric fluid and a return to normal tubular morphology of the body column (*Wanek et al., 1980*). Thus, one function of nerve cells in the endoderm is the control of endodermal muscle tension in response to osmotic pressure.

## A role for the endodermal nerve net in feeding?

A striking feature of the endodermal nerve net is the large proportion of sensory cells, which constitute >50% of all endodermal nerve cells. These cells are oriented toward the gastric cavity (*Figure 2B*) and have sensory cilia, which extend into the gastric cavity to monitor digestive activity and perhaps also osmotic strength. They are connected to the ganglion network at their basal ends and could induce signals affecting changes in endodermal muscle processes associated with digestion. Two such events occur following ingestion of prey. First endodermal muscle processes below the tentacle ring contract locally ('neck formation') while endodermal muscles in the body column relax (*Blanquet and Lenhoff, 1968*). Together these events create a closed space in the gastric cavity for digestion of prey. After digestion is complete, slow contraction of endodermal muscle processes restores the normal columnar form of the body column.

Recent experiments of *Giez et al., 2023a*, have now shown that endodermal ganglion cells are responsible for this regulation. These N4 nerve cells fire in response to swelling of the body column, e.g., when a food pellet is present in the gut. Ablation of N4 nerve cells using the nitroreductase-metronidazole system (*Curado et al., 2007*) prevents mouth opening and ejection of the food pellet, which leads to dramatic bursting of the gut. Expulsion of glass beads used as an artificial food pellet was also significantly delayed in N4 ablated animals. Hence, endodermal ganglion cells are involved in regulating mouth opening and the swelling of the body column. The latter is a slow process, which could be caused by diffuse release of neuronal signaling molecules affecting muscle tension in endodermal muscle cells.

During digestion, slow waves of radial contraction move along the oral/aboral axis mixing the contents of the gut (*Shimizu et al., 2004*). Both neck formation and radial contractions are absent or strongly reduced in nerve-free animals consistent with a role for the endodermal nerve net in regulating digestion. The complete absence of nerve cells in the endoderm of the tentacles (*Figure 5*), which are not involved in digestion, is consistent with this conclusion.

## A model for continuous expansion of the nerve net in growing *Hydra* polyps

*Hydra* polyps grow continuously, but maintain a nearly constant size. The increasing cell numbers are displaced into buds, which develop in the lower body column and into the tentacles and the basal disk, which are terminally differentiated and continuously replaced by tissue from the body column (*Campbell, 1967*). As a consequence of this tissue growth, the *Hydra* nerve net undergoes continuous addition of new cells. This addition has been clearly shown in experiments tracking the differentiation of DiI-labeled nerve precursors in live animals (*Hager and David, 1997*) or immigration of Brdu-labeled precursors (*Technau and Holstein, 1996*). Nerve cell differentiation occurred throughout the body column and maintained the constant ratio of nerve cells to epithelial cells (*Figure 6*).

The DiI experiments, however, could not visualize how newly differentiated nerve cells attached to the nerve net, since the nerve net was not labeled. The new experiment shown in *Figure 13*, in which the nerve net was imaged with the PNab, demonstrates that new nerve cells are added laterally to the existing nerve net. Thus, the nerve net is an overlay of individual nerve cells and neurites as shown in *Figures 7 and 8* and this overlay is a consequence of how it grows. *Figure 14* summarizes these results in a simple model for growth of the nerve net. One consequence of this model is that elongating neurites terminate at apparently unspecific positions along a pre-existing neurite track (*Figures 7 and 10*). A second consequence is that elongating neurites grow past neighboring nerve

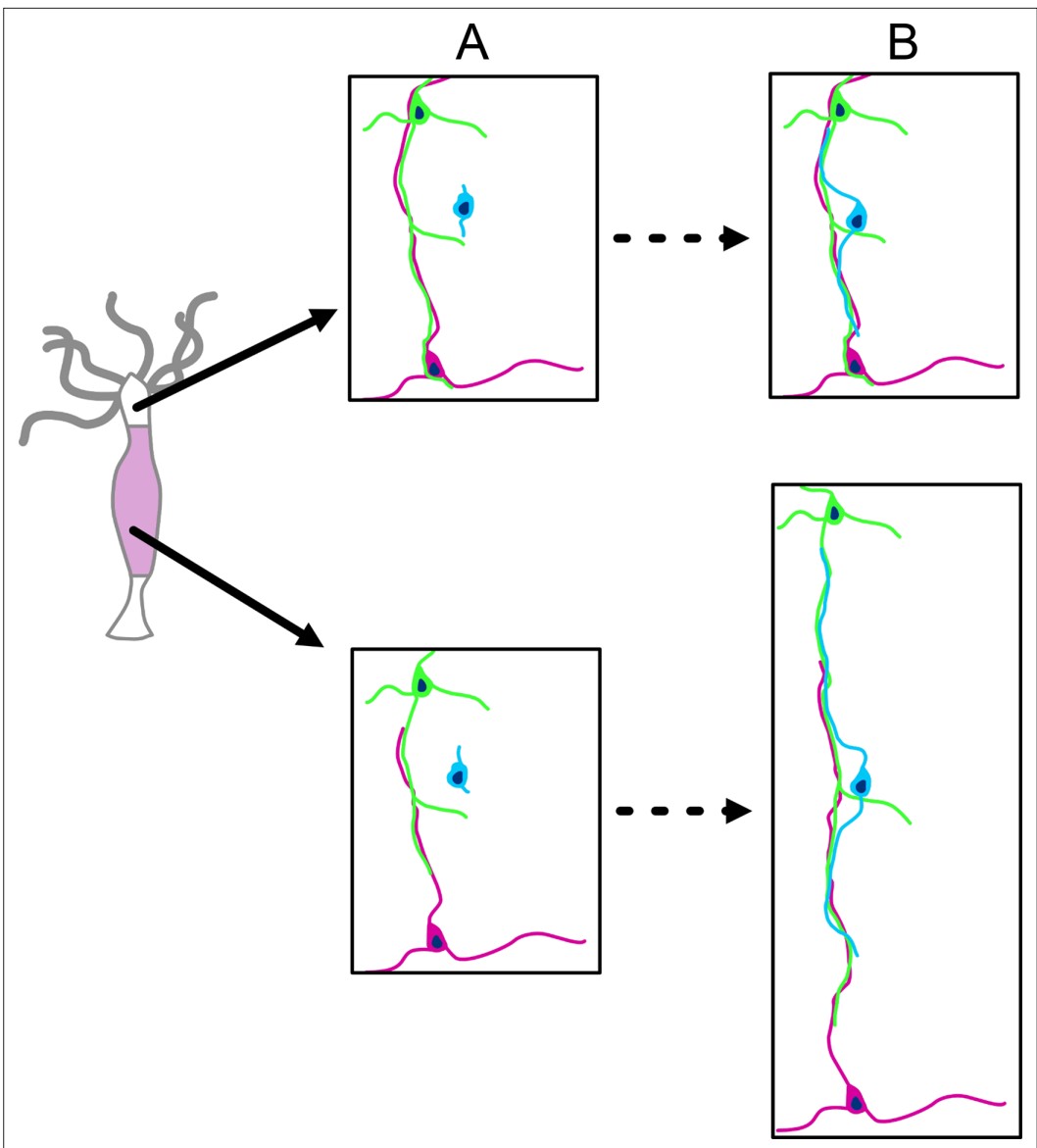

**Figure 14.** Schematic diagram of nerve net growth by lateral association of newly differentiated nerve cells (blue) with the pre-existing nerve net (green and red). Upper images: the base of tentacles; lower images: the middle of the body column. The images are approximately to scale and show changes associated with tissue growth. (**A**) Nerve net at $t_0$. (**B**) Nerve net during tissue growth. Lower image pair: new nerve cell (blue) attaches laterally to the nerve net (as shown in *Figure 13*) and thus maintains the ratio of nerve cells to epithelial cells during growth of the nerve net. Upper image pair: new nerve cell attaches laterally to the nerve net at the base of a tentacle. The nerve net does not grow at this position but is displaced into the tentacle. Newly differentiated nerve cells thus increase the ratio of nerve cells to epithelial cells in the ectoderm of the tentacle (see *Figure 6*).

cell bodies (*Figures 9A and 10*). Both features are supported by our results and suggest that the existing nerve net acts as a template for the addition of new nerve cells.

## Nerve cell differentiation in the hypostome, tentacles, and peduncle

The results in *Figure 6* show that the nerve cell to epithelial cell ratio in the endoderm is nearly constant along the body column. As discussed above, this ratio can be maintained by continuous addition of new cells to the nerve net during growth. Similarly, in the ectoderm of the gastric region and the peduncle, the nearly constant ratio of nerve cells to epithelial cells can be maintained by addition of new cells to the existing nerve net.

The tentacles, hypostome, and basal disk, however, all contain increased numbers of nerve cells in the ectoderm (*Figure 6*) compared to the body column tissue from which they arise by tissue displacement. This implies new nerve cell differentiation in these tissues. This new differentiation has been well documented in earlier experiments (*David and Gierer, 1974*; *Hager and David, 1997*; *Hobmayer et al., 1990a*; *Hobmayer et al., 1990b*; *Technau and Holstein, 1996*; *Yaross and Bode, 1978a*; *Yaross and Bode, 1978b*) and is presumably associated with additional sensory and motor activities in these tissues. Indeed, a fourth GCaMP6s activity pattern was identified in nerve cells at the base of tentacles (*Dupre and Yuste, 2017*), which was associated with movement of the head (nodding). A specific population of sensory nerve cells (Nv1) was identified which are incorporated into the battery cell complex in the ectoderm of tentacles (*Hobmayer et al., 1990a*; *Hobmayer et al., 1990b*). At the opposite end, in the lower peduncle, a population of nerve cells was identified, which express the *innexin2* gene (*Takaku et al., 2014*). Elimination of the activity of these gap junctions disrupted contractile activity of the body column. In all of these examples, it is possible to imagine that the lateral addition of new nerve cells to the existing nerve net (see *Figure 14*) gives rise to local changes in neural activity.

Finally, the nerve net is not just modified by addition as described above, but also by deletion of nerve cells. As shown in *Figures 5 and 6*, there are no nerve cells in the endoderm of the tentacles, even though this tissue arises by displacement of body column tissue, which contains nerve cells in the endoderm. This implies that nerve cells in the endoderm are removed when endodermal tissue moves into tentacles. It appears likely that this occurs by apoptosis (*Cikala et al., 1999*), although this has not been observed directly. A similar process presumably removes nerve cells from endodermal tissue entering the basal disk. It also indicates a constant turnover of existing neuron-neuron and neuron-epithelial cell interactions. The molecules involved in the underlying feedback communication for these processes are currently unknown.

## Materials and methods
### Animals
*Hydra* strains were maintained in laboratory cultures and fed two to three times per week with *Artemia* nauplii. *Hydra* medium (HM) contains 1 mM NaCl, 1 mM CaCl$_2$, 0.1 mM KCl, 0.1 mM MgSO$_4$, 1 mM Tris (pH 7.6). The following strains were used for all experiments: *Hydra vulgaris* AEP (wildtype); transgenic strains: watermelon (expressing GFP and DsRed2 in ectodermal and endodermal epithelial cells respectively); inverse watermelon (expressing DsRed2 and GFP in ectodermal and endodermal epithelial cells respectively; *Glauber et al., 2013*); nGreen (expressing GFP in nerve cells, a few nests of differentiating nematocytes and gland cells; see below); Lifeact ecto (expressing Lifeact peptide-GFP fusion protein in ectodermal epithelial cells); Lifeact endo (expressing Lifeact peptide-GFP in endodermal epithelial cells; *Aufschnaiter et al., 2017*); Hym176B (expressing GFP in a subpopulation of ectodermal nerve cells under control of the Hym176B promoter; *Noro et al., 2019*); alpha-tubulin pan-neuronal (expressing NeonGreen in all nerve cells under control of an alpha-tubulin promoter; *Primack et al., 2023*).

### Generation of the nGreen transgenic line
The nGreen line was generated by microinjection of embryos from the SS1 strain of *H. vulgaris* AEP. SS1 is an F1 line that was generated by self-crossing of AEP. The plasmid pHyVec1 (GenBank accession number AY561434), which contains an actin promoter driving expression of GFP, was used for injection. Injection and screening of embryos was carried out as previously described (*Wittlieb et al., 2006*; *Juliano et al., 2014*). The nGreen line showed GFP expression in all types of nerve cells, a few nests of differentiating nematocytes and gland cells. The nGreen line used here is a mosaic containing both transgenic and non-transgenic interstitial stem cells. As a consequence, not all differentiated cells express GFP.

### Counting cells by tissue maceration
Tissue maceration to identify and count *Hydra* cell types was carried out as previously described (*David, 1973*; *Bode et al., 1973*) with modifications. Maceration solution is composed of glycerol, glacial acetic acid, and water (1:1:13). Whole *Hydra* or tissue pieces were incubated in maceration

solution for 10 min with gentle shaking to dissociate the tissue. An equal volume of 8% paraformal-dehyde (PFA) was added to fix the macerated cells. 50 µl of the suspension of macerated cells were spread evenly over about 2 cm² on a subbed microscope slide and dried. Two drops of glycerol:water (1:1) were added to the dried spread and covered with a coverslip. Nerve cells and epithelial cells were counted under a phase contrast microscope.

## Counting nerve cells and epithelial cells in confocal image stacks

To count ectodermal and endodermal epithelial cells in confocal stacks, we used transgenic animals expressing GFP in the ectoderm and DsRed2 in the endoderm (watermelon line) or vice versa (inverse watermelon line). Since both DsRed2 and GFP are partially concentrated in the nuclei of transgenic epithelial cells, nuclei of epithelial cells can be distinguished from the DAPI-stained nuclei of other cells. GFP nuclei are turquoise and DsRed2 nuclei are magenta (see *Figure 1—figure supplement 1*). The nuclei of all other cells are blue (DAPI). *Figure 1—figure supplement 2B* shows a confocal image stack in the ectoderm of a transgenic watermelon *Hydra* expressing GFP in ectodermal epithelial cells. The epithelial cell nuclei are clearly visible as round spots lying above the ectodermal muscle fibers oriented parallel to the body axis. Nerve cells stained with PNab in the same image stack are shown in *Figure 1—figure supplement 2A*. Yellow spots indicate nerve cell nuclei confirmed by DAPI staining.

## Preparation and specificity of the PNab antibody

A rabbit antibody was generated against the peptide sequence VTRNQQDQQENRFSNQ, which corresponds to amino acids 4372–4388 in the cytoplasmic domain of the classic cadherin of *H. vulgaris (magnipapillata strain 105)* (GenBank accession number XP_012558722). Peptide synthesis, immunization of three rabbits, and subsequent isolation and purification of the IgG fraction were carried out by Dr. J Pineda Antibody Service (Berlin). The resulting antibody was highly specific for nerve cells in *H. vulgaris* (*Cramer von Laue, 2003*; *Bertulat, 2008*). Pre-absorption with the synthetic peptide blocked antibody staining (*Cramer von Laue, 2003*).

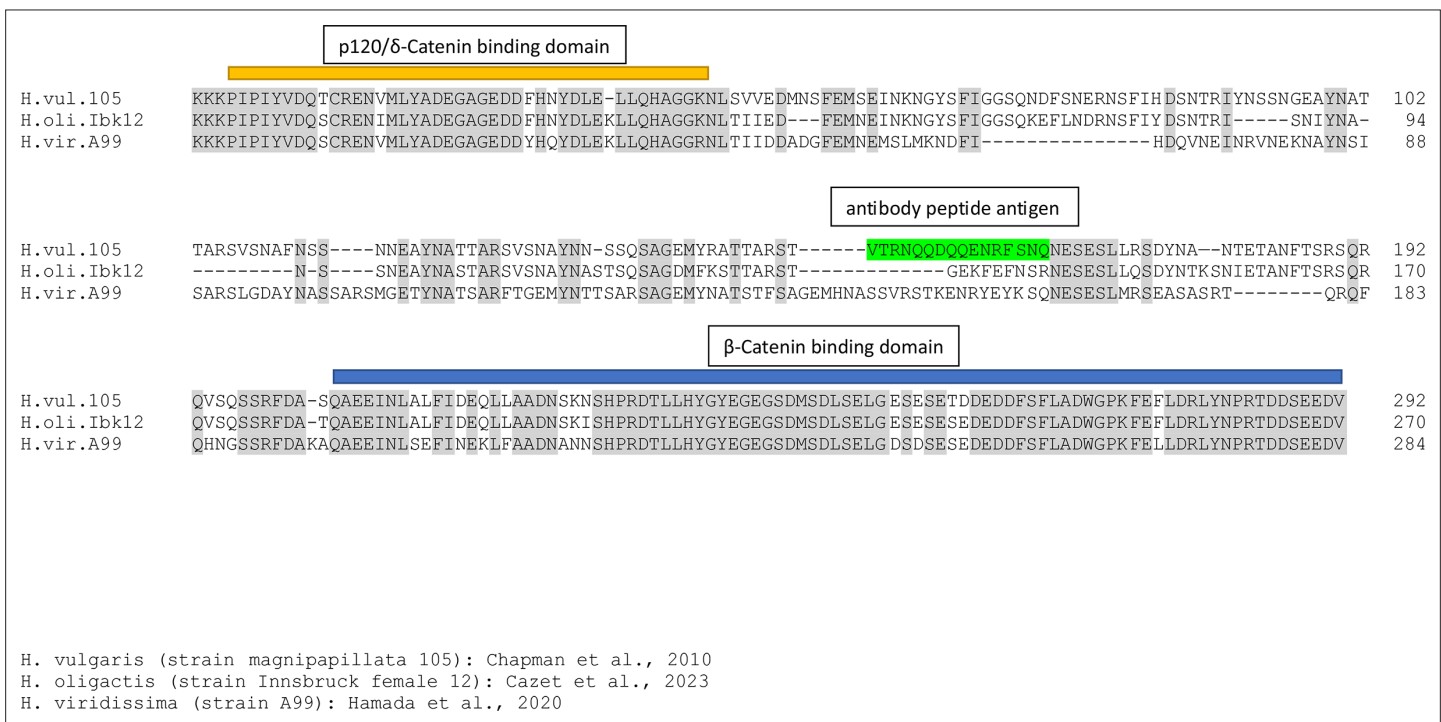

**Figure 15.** Sequence alignment of the intracellular domains of classic cadherin proteins from *H. vulgaris* (strain 105), *H. oligactis*, and *H. viridissima*. The p120/delta-catenin and the beta-catenin binding domains are highly conserved in all three species and anchor the alignment. The sequence between these two conserved domains is quite variable and, in particular, the 18 amino acid sequence (green) used to make the pan-neuronal antibody (PNab) is only present in the *H. vulgaris* sequence.

Although PNab was made to a peptide in the classic cadherin of *H. vulgaris*, it is now clear that it does not recognize the cadherin protein. Both *H. oligactis* and *Hydra viridissima* have classic cadherin proteins and nerve cells in both of these species also stain strongly with PNab (*Holstein, 2023*). However, newly available cadherin gene sequences from *H. oligactis* and *H. viridissima* show that the sequences of the intracellular domain of classic cadherin differ dramatically between the three species (*Figure 15*). In particular, the 18 amino acid sequence used to make the PNab antibody is only present in the *H. vulgaris* cadherin but not in the *H. oligactis* and *H. viridissima* cadherins. Hence, it is clear that PNab cannot be staining cadherin in neurons and the identity of the PNab antigen is presently unknown. Nevertheless PNab is a highly specific reagent to identify nerve cells in all species of *Hydra*.

## Immunofluorescent staining of fixed *Hydra*

Immunofluorescent staining was performed based on a protocol optimized for double-labeling experiments with other polyclonal antibodies (*Bertulat, 2008*). Prior to fixation, animals were relaxed with 2% urethane in HM for a few minutes to prevent excessive contraction induced by fixation. Animals were fixed for 1 hr in 4% PFA in PBS (120 mM NaCl, 50 mM $K_2HPO_4$, 12 mM $NaH_2PO_4$, pH 7.2–7.4). Following fixation animals were washed 3× in PBS. Animals used in *Figure 8* were fixed with Lavdowsky's fixative (48% ethanol, 3.6% formaldehyde, and 3.8% acetic acid in water).

To improve antibody access to all cells in the polyp, fixed animals were cut in half or into smaller pieces depending on the experiment. Cutting animals also enabled better mounting of body parts for imaging, e.g., top view of the hypostome or horizontal rings of the body column allowing views down the y-axis. Cutting was performed after fixation in order to maintain the structural features of the tissue, which was important for analysis of neuronal network characteristics.

Immunostaining was performed according to the following protocol. Fixed tissue was washed 3× in PBS, permeabilized in 0.5% Triton in PBS for 15 min, blocked for 20 min in 10 mg/ml bovine serum albumin, 0.1% Triton in PBS. Tissue pieces were incubated in primary antibody in blocking solution overnight at 4°C, then washed 3× in PBS, and incubated in secondary antibody in blocking solution for 2 hr. Tissue fragments were then washed 3× in PBS (10 min), stained with DAPI (1 mg/ml) in PBS (10 min), and washed 1× in PBS.

Stained specimens were mounted on microscope slides with VECTASHIELD HardSet Antifade Mounting Medium. Bee wax was used on the edges of coverslip to prevent squashing of the preparations. Coverslips were sealed with nail polish and slides were stored at 4°C.

Primary and secondary antibodies used for staining, together with their respective dilutions, are listed below:

Primary antibodies: PNab (rabbit) 1/1000 (Thomas Holstein, Heidelberg); anti-GFP (chicken) 1/500 (Aves Lab).

Secondary antibodies: Alexa 488 donkey anti-chick 1/400 (Dianova 703-546-155); Cy3 donkey anti-mouse 1/200 (Dianova 715-166-151); Cy3 donkey anti-rabbit 1/400 (Dianova 711-165-152).

All immunostaining experiments were repeated multiple times (technical replicates). Usually two to five animals were used per immunostaining (biological replicates). All stained animals showed the same results upon imaging.

## Confocal imaging

Specimens were examined with a Leica SP5 scanning confocal microscope. Image stacks were taken with ×10, ×20, and ×63 objective lenses. Images at ×10 and ×20 magnification were taken every 1 μm. Images at ×63 magnification were taken every 0.3 μm. Images were processed with ImageJ. The plugin StackGroom was used to correct z-shift and to create RGB stacks. Most images shown are maximum intensity projections of short stacks (5–15 μm).

## TEM and SBFI

Sample preparation was carried out according to a modified protocol of *Deerinck et al., 2010*, adapted for *Hydra*. In brief, *Hydra* polyps were relaxed with 2% urethane in HM and immediately fixed with 2.5% glutaraldehyde in HM for 5 min, cut in half and further fixed for 2 hr. After washing in cacodylate buffer, samples were postfixed with ferrocyanide osmium tetroxide, treated with thiohydrocarbazide-osmium, and in block stained with uranyl acetate and lead aspartate to enhance contrast. After dehydration with acetone, samples were infiltrated with increasing concentrations of acetone-resin and

finally embedded in Durcupan ACM resin, mixture hard. The samples were oriented in small silicon molds and polymerized at 60°C for 48 hr.

For TEM, ultra-thin sections were cut with an ultramicrotome UCT (Leica, Vienna) using a diamond knife (Diatome, Switzerland) and examined with an Energy Filter Transmission Electron Microscope Libra 120 (Zeiss, Germany). Images were acquired using a 2×2k high-speed camera and an ImageSP software (Tröndle, Germany).

For serial block face imaging (SBFI), the resin block was mounted on a SEM stub with conductive glue, trimmed to 0.5×0.5×0.5 mm$^3$ and coated with gold with a sputter coater. 3D imaging was performed on an FEI Quanta FEG 250 scanning electron microscope with an integrated ultramicrotome (prototype, workgroup W Denk, MPI Neurobiology Munich) at 2.5 kV. 2000 planes were cut in 30 nm steps at a lateral resolution of 20×20 nm$^2$, resulting in a volume of 40×13×56 µm$^3$. Neurites, nerve cell bodies, and nuclei were segmented and rendered using Amira 5.6 software.

TEM imaging was carried out on multiple animals (>20) (*Figure 9*). Block face SEM was done on one animal (*Figure 10*).

## Acknowledgements

The authors thank Shawn Mikula of the Denk Lab at the Max Planck Institute for Neurobiology in Martinsried for help with block face sectioning, the Center for Advanced Light Microscopy (CALM) in the Biozentrum for access to confocal microscopes, Angelika Boettger for laboratory space and resources, Sigrid Grieser-Ade, Natalie Kolb, and Anne Kuhn for *Hydra* culture and Hermann Rohrer and Suat Özbek for discussions. The images in *Figure 11* were generously provided by Amber Louwagie.

## Additional information

### Funding

| Funder | Grant reference number | Author |
| --- | --- | --- |
| Marie Skłodowska-Curie Actions | 10.3030/847681 | Bert Hobmayer |
| Deutsche Forschungsgemeinschaft | Ho1621/3 | Thomas W Holstein Bert Hobmayer |
| Deutsche Forschungsgemeinschaft | SFB474/B1 | Thomas W Holstein Bert Hobmayer |
| Deutsche Forschungsgemeinschaft | Ho1088/2 | Thomas W Holstein |
| Deutsche Forschungsgemeinschaft | SFB 269/A5 | Thomas W Holstein |
| Deutsche Forschungsgemeinschaft | SFB 488/A12 | Thomas W Holstein |
| Deutsche Forschungsgemeinschaft | SFB 873/A1 | Thomas W Holstein |

The funders had no role in study design, data collection and interpretation, or the decision to submit the work for publication.

### Author contributions

Athina Keramidioti, Sandra Schneid, Christina Busse, Validation, Investigation, Visualization, Methodology, Writing – review and editing; Christoph Cramer von Laue, Bianca Bertulat, Kristine M Glauber, Resources, Methodology; Willi Salvenmoser, Olga Alexandrova, Investigation, Visualization, Methodology; Martin Hess, Investigation, Visualization, Methodology, Writing – review and editing; Robert E Steele, Resources, Investigation, Writing – review and editing; Bert Hobmayer, Conceptualization, Formal analysis, Supervision, Funding acquisition, Project administration, Writing – review and editing; Thomas W Holstein, Conceptualization, Resources, Formal analysis, Supervision, Funding acquisition, Project administration, Writing – review and editing; Charles N David, Conceptualization, Data

curation, Formal analysis, Supervision, Writing – original draft, Project administration, Writing – review and editing

### Author ORCIDs
Athina Keramidioti ⓘ http://orcid.org/0009-0005-7209-3028
Kristine M Glauber ⓘ http://orcid.org/0000-0002-1093-6481
Robert E Steele ⓘ https://orcid.org/0000-0001-5355-1656
Thomas W Holstein ⓘ https://orcid.org/0000-0003-0480-4674
Charles N David ⓘ https://orcid.org/0000-0002-9811-4043

Reviewer #2 (Public Review): https://doi.org/10.7554/eLife.87330.3.sa1
Reviewer #3 (Public Review): https://doi.org/10.7554/eLife.87330.3.sa2
Author Response https://doi.org/10.7554/eLife.87330.3.sa3

## Additional files

### Supplementary files
• MDAR checklist

### Data availability
Hydra strains used in our experiments are freely available on request from Rob Steele (UC Irvine), Thomas Holstein (Heidelberg University), Bert Hobmayer (University of Innsbruck) and Celina Juliano (UC Davis). The pan-neuronal antibody PNab is available from Thomas Holstein (Heidelberg University).

The following previously published dataset was used:

| Author(s) | Year | Dataset title | Dataset URL | Database and Identifier |
|---|---|---|---|---|
| Cazet JF, Siebert S, Little HM, Bertemes P, Primack AS, Ladurner P, Achrainer M, Fredriksen MT, Moreland RT, Singh S, Zhang S, Wolfsberg TG, Schnitzler CE, Baxevanis AD, Simakov O, Hobmayer B, Juliano CE | 2023 | Data from: A chromosome-scale epigenetic map of the Hydra genome reveals conserved regulators of cell state | https://biowebprod22.nhgri.nih.gov/HydraAEP/SingleCellBrowser/ | Hydra AEP Genome Project Portal, HydraAEP/SingleCellBrowser |

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
