## [Editor Report · eLife assessment]

This work presents **important** findings on the cellular and ultrastructural organization of the nervous system in the freshwater polyp Hydra. The authors present outstanding imaging data with **convincing** evidence to support their claims. The manuscript provides a starting point for further functional in vivo studies. The work will be of interest to developmental biologists and neurobiologists.

---

## [Referee Report · Reviewer #2 (Public Review)]

In their manuscript, Keramidioti and co-authors investigate the cellular architecture of the nervous system in the freshwater polyp Hydra. Specifically, the authors attempt to improve the resolution, which is lacking in the previous studies, yet to generate a comprehensive overview of the entire nervous system's spatial organization and to infer communication between cells. To this end, Keramidioti et al. use state-of-the-art imaging approaches, such as confocal microscopy combined with the use of transgenic animals, transmission electron microscopy, and block face scanning electron microscopy. The authors present three major observations: (i) A novel PNab antibody may be used to detect the entire nervous system of Hydra; (ii) Nerve cells in the ectoderm and in the endoderm are organized in two separate nerve nets, which do not interact; (iii) Both nerve nets are composed of bundles of overlapping nerve processes.

The manuscript addresses a long-standing and currently intensively studied question in developmental neurobiology biology - it attempts to reveal structural properties and principles that govern the function of the nervous systems in non-bilaterian animals. Hence, this study contributes to understanding the nervous system evolution trajectories. Therefore, the manuscript may represent interest to researchers interested in evolutionary and developmental neurobiology.

The manuscript reports a remarkably meticulous study and presents stunning imaging results.

---

## [Referee Report · Reviewer #3 (Public Review)]

In this paper by Keramidioti et al, the authors have characterized a polyclonal antibody from rabbit, which was raised against a peptide of the intracellular domain of the Hydra Cadherin. This antibody unexpectedly recognizes presumably all neurons in the Hydra polyp but the specificity of the antibody was not investigated. Regardless, the antibody can be used to visualize and study the nerve net under a variety of conditions. The authors find that the endodermal and ectodermal nerve net do not make any contacts through the mesoglea, in contrast to earlier assumptions and data. They show that ectodermal neurons make close contacts to the myoepithelial muscles, in contrast to the endodermal muscles. Furthermore, they show that tentacle endoderm surprisingly does not have any neurons. Finally, a very nice tool to visualize the connections between the neurons is the staining of mosaic nGreen transgenic lines. This showed that the neurites align in parallel forming bundles of neurites over longer stretches, in particular in the ectoderm, which offers a mechanism how new neurons are added laterally to the existing nerve net. This has important implications about the way the neurons might communicate with each other.

Taken together, this paper adds to our knowledge of the Hydra nerve net and provides a new experimental tool. Although most of the study is rather descriptive the pictures are of spectacular quality, providing fascinating new insights into the arrangement and topology of the nerve net.

---

## [Author Response]

The following is the authors’ response to the original reviews.

Answers to reviewers’ comments

Peer Reviewers 2 and 3 criticized the name of the antibody – hvCADab - and the lack of proof that it recognized a classic cadherin. These criticisms were justified and in the intervening months the issue has been resolved. hvCADab does not recognize the cadherin protein, although it was made to an 18 amino acid sequence from the intracellular domain of the H. vulgaris cadherin protein. Newly available genome sequences from two other species, Hydra oligactis and Hydra viridissima, now show that the 18 amino acid antigen sequence is not present in these species.

Nonetheless, the nerve net in both species is strongly stained by the antibody. Hence we have renamed the antibody PNab (pan-neuronal antibody). The antigen is currently not known. Nevertheless the antibody is an excellent reagent for imaging the nerve net in Hydra.

We have revised the section on antibody preparation in Materials and Methods to state explicitly that PNab does not recognize classic cadherin. To support this conclusion we have added a sequence comparison (Suppl Fig 3) of the intracellular domains of classic cadherins from H. vulgaris, H. oligactis and H. viridissima, which show that the 18aa antigen sequence is only present in the H. vulgaris classic cadherin and not in the cadherin sequences from H. oligactis and H. viridissima. All three sequences have highly conserved p120/delta-catenin and beta-catenin binding domains. The sequence between these domains is highly variable and the 18aa antigen sequence used for antibody production is clearly not present in the H. oligactis and H. viridissima sequences.

Both reviewers also criticized our evidence for pan-neuronal staining as inadequate. Hence we have now included additional data. We have stained a transgenic strain expressing NeonGreen under the control of a pan-neuronal alpha-tubulin promoter (Primak et al 2023). 684/684 transgenic nerve cells were stained with PNab. We consider this convincing evidence, in addition to the evidence presented previously, that PNab stains all nerve cells in Hydra. The first paragraph of Results has been revised to include these data.

Reviewer 2 suggested moving gap junction/innexin data (Suppl Fig 3 and 4) from the Discussion to Results. These are indeed new results and we have followed this suggestion. Fig 12 (new) clearly shows gap junctions between neurites in bundles. It also shows that nerve cells in bundles express cell type specific innexins and hence can form cell type specific gap junctions. We have also added new images (Fig 11) of a transgenic Hym176B strain stained with PNab. These show that neurite bundles in the ectoderm contain neurites from different nerve cell types = neural circuits and hence that neurite links must be specific, e.g. gap junctions.

As suggested by Reviewer 2 we have now provided a 3D interactive version of the block face SEM reconstruction (Suppl Fig 4). This shows that connections between neurites in bundles consist of thin overlapping fingers rather than “conventional” terminal contacts. It also shows that the purple neurite and extends past the green nerve cell body and does not end on it.

Reviewer 2 suggested deleting discussion of possible functions for the endodermal nerve net (Discussion). We disagree with this suggestion. Our imaging results showed no connections between ectodermal and endodermal nerve nets. We also presented quantitative data for the absence of contact between the nerve nets in the gastric region. Consistent with our observations, Dupre and Yuste (2017) found no functional connection between the ectodermal and endodermal nerve nets based of neural activity measurements. Nevertheless, Giez et al (2023) in a recent preprint have described contact between specific endodermal and ectodermal nerve cells in the hypostome involved in the mouth opening response to glutathione. Both their observation and ours may be correct. The issue is not resolved. Hence we have included a discussion of possible functions for ectodermal and endodermal nerve nets. Importantly, our conclusions incorporate the difference in connectivity between muscle processes and nerve cells in the two nerve nets.

Specific comments / Recommendations

**Reviewer 2**

Novelty: two preprints (Giez et al 2023) became available after the submission of our preprint. These include the results cited by the reviewer. These were not available to us at the time of submission.

hvCADab has been re-named (see above). The differentiating nerve cell in Fig 11B is indeed stained by PNab. We have adjusted the intensities of red and green channels to show this more clearly.

We consider the very clear black space between ectoderm and endoderm e.g. Fig 2B or Fig 4A to be an adequate marker for mesoglea. Use of an anti-mesoglea antibody would reduce the clarity of the image.

It is always possible to look at more parts of Hydra tissue for possible nerve connections between ectoderm/endoderm. Nevertheless we provide the first quantitative data on the lack of contacts between 133 nerve cells (57 ectodermal and 76 endodermal) in the body column. Such data has not been previously available. And the EM result (Westfall 1973) cited by the reviewer is anecdotal at best. In later serial sectioning results on the hypostome/tentacle region from the Westfall lab no mention is made of nerve connections between the ectoderm and the endoderm. However, based on the results in the cited preprints (Giez et al) a closer examination of the hypostome/tentacle region in particular is warranted.

To strengthen our conclusion that there are no contacts between the ectodermal and endodermal nerve nets, we now explicitly cite results from Dupre and Yuste (2017) on a calcium reporter strain demonstrating the absence of any crosscorrelation between the firing patterns of ectodermal RP1 network and the endodermal RP2 network. There was also no correlation between the activity of the second ectodermal nerve net CB and the endodermal RP2 network. These results demonstrate the absence of functional contacts between ectodermal and endodermal nerve nets.

The reviewer criticizes the absence of trans-mesoglea links between ectodermal and endodermal epithelial cells in our EM images, e.g. Fig 9A. We can assure the reviewer that such links are frequently observed, although not in the image we chose for Fig 9A. This image, however, clearly documents two neurite bundles next to ectodermal muscle fibers.

We agree with the reviewer that neurite bundles are an important discovery. And they raise the question of synaptic connections between neurites in bundles. Unfortunately, it is not possible to scan along the block face reconstruction (Fig 10) and count synapses. The resolution is not sufficient. Although scattered dense core vesicles (DCV) are observed in neurites, clustered DCV described by Westfall et al (1971) as synapses were not observed. We did, however, observe gap junctions between neurites in bundles (noted in Suppl Fig 3). These data have now been moved to the main body of the paper as Fig 12 together with the scRNAseq results on innexin gene expression in nerve cells. These results make it clear that neurites in bundles are connected via gap junctions and that these gap junctions are specific for neural circuits.

The reviewer suggests that neurite bundles are an artifact of their interaction with muscle processes at the base of epithelial cells. We disagree with this statement. Muscle processes are temporary structures. They are withdrawn and reformed during every epithelial cell division, which occur approximately every three days. Bundles are almost certainly more stable structures. Furthermore, neurite bundles in the endoderm are distant from endodermal muscle fibers (Fig 4B and Fig 9D) and their polygonal pattern (Fig 2D) is completely different from the circumferential bands of endodermal muscle fibers.

**Reviewer 3**

Specific comments and suggestions have been answered above. Importantly, we show that the PNab antibody does not recognize cadherin and that it clearly stains all nerve cells in Hydra.